# Spectrometric measurements of atmospheric propane ($C_3H_8$)

Geoffrey C. Toon[1], Jean-Francois L. Blavier[1], Keeyoon Sung[1], Katelyn Yu[1,2]

[1] Jet Propulsion Laboratory, California Institute of Technology, Pasadena, CA, 91109, USA

[2] Dept. Civil and Environmental Engineering, UC Berkeley, Berkeley, CA, 94720, USA

*Correspondence to*: Geoffrey.C.Toon@jpl.nasa.gov

**Abstract.** We report measurements of atmospheric $C_3H_8$ from analysis of ground-based, solar absorption spectra from the JPL MkIV interferometer. Using the strong Q-branch absorption feature at 2967 cm$^{-1}$, we can measure $C_3H_8$ in locations where its abundance is enhanced by proximity to sources (e.g., large natural gas fields, mega-cities). A case study of MkIV $C_3H_8$ measurements from Ft. Sumner, New Mexico, shows that amounts are strongly correlated with ethane ($C_2H_6$) and with back-trajectories from SE New Mexico and West Texas, where the Permian Basin oil and natural gas field is located. Measurements from JPL, California, also show large $C_3H_8$ enhancements on certain days, but more correlated with CO than $C_2H_6$. From high-altitude, balloon-borne, MkIV solar occultation measurements, $C_3H_8$ was not detected at any altitude (5-40 km) in any of its 25 flights.

## 1. Introduction

Non-methane hydrocarbons such as $C_3H_8$ and $C_2H_6$ affect air quality because their oxidation enhances tropospheric $O_3$ and aerosol pollution. They are also sensitive indicators of fugitive losses by the oil and natural gas industry, an important source of co-emitted methane ($CH_4$), a greenhouse gas. These fugitive losses appear to be under-estimated in global inventories (Dalsoren et al., 2018).

Atmospheric $C_3H_8$ and $C_2H_6$ are entirely the result of emissions at the surface. In pre-industrial times these came from geological seeps and wild fires, but in recent times these natural sources have been surpassed by emissions from fossil fuel production. The latter peaked in about 1970, and then declined due to stricter regulation of emissions from the oil and natural gas industry and automobiles. But in the past decade, this decreasing trend has reversed due to accelerated Natural Gas (NG) exploitation (Helmig et al., 2016).

$C_3H_8$ has a lifetime of about 2 weeks in summer and 8 weeks in winter (Rosado-Reyes et al., 2007). This is mostly dictated by how fast it is being oxidized by reactions with hydroxyl radicals and chlorine atoms. Given this 2–8 week lifetime, a single strong source of propane has the potential to degrade air quality over most of the hemisphere.

Unprocessed, in-the-ground, "wet" natural gas is usually between 70–95% $CH_4$, 1–15% $C_2H_6$, 1–10% $C_3H_8$, and 0–3% $C_4H_{10}$. The latter two gases are typically extracted to form Liquified Petroleum Gas (LPG). In the northern hemisphere winter, LPG contains more $C_3H_8$, while in summer it contains more butane ($C_4H_{10}$), reducing variations in its vapor pressure.

---

Deleted: large variations

Deleted: amounts

Formatted: Font: Times New Roman, 10 pt

Deleted: From MkIV solar occultation measurements from balloon, $C_3H_8$ was not detected at any altitude in any flight

Formatted: Font: Times New Roman, 10 pt, Subscript

Formatted: Font: Times New Roman, 10 pt

Formatted: Font: Times New Roman, 10 pt

Formatted: Font: Times New Roman, 10 pt, Subscript

Formatted: Font: Times New Roman, 10 pt

Moved down [1]: In contrast, the lifetime of $C_2H_6$ is 2–6 months, which is 3–4 times longer than that of $C_3H_8$.

Moved (insertion) [1]

Deleted: In contrast, the lifetime of $C_2H_6$ is 2–6 months, which is 3–4 times longer than that of $C_3H_8$.

LPG burns much more cleanly than fuel oil and is therefore increasingly used for heating, and cooking, especially in
rural areas that are not served by piped NG. LPG is also used to fuel commercial vehicles, and is increasingly
replacing CFCs as a refrigerant and as an aerosol propellant. As a result of extracting LPG from natural gas, the NG
that is piped to our homes in urban areas is highly depleted in $C_3H_8$ and $C_4H_{10}$, as compared with wet NG.
To the best of our knowledge, there are no previous remote sensing measurements of $C_3H_8$, although in situ
measurements exist. Dalsoren et al. (2018, Fig.3b) show surface in situ $C_3H_8$ amounts below 50 ppt at Zeppelin
station in Svalbad in summer 2011, but with values of 1 ppb in the winter, with peaks of up to 2.4 ppb. These $C_3H_8$
peaks are strongly correlated with $C_2H_6$ which reaches 3.4 ppb. Using in situ $C_3H_8$ data from multiple sites Helmig et
al. (2016) show a large seasonal cycle in surface in situ $C_3H_8$ at high NH latitudes, reaching 1 ppb in winter, with
little in the SH. They also show increasing $C_3H_8$ over central and Eastern US over the period 2009.5–2014.5, but no
increase on the West coast.
Since $C_3H_8$ correlates with $C_2H_6$, both having NG as their main source, we also consider the previous measurements
of $C_2H_6$, which has a lifetime of 2-8 months, 4 times longer than propane. Angelbratt et al. (2011) reported a 0–
2%/year decline over the period 1996 to 2006 based on data from six NH FTIR sites. Franco et al (2015) reported a
shallow minimum in $C_2H_6$ in the 2005–2010 period based on ground-based FTIR solar spectra above the
Jungfraujoch scientific station. Helmig et al. (2016) report a minimum in atmospheric $C_2H_6$ in 2005–2010 based on
in situ and remote measurements.
Franco et al. (2016) estimate a 75% increase in North American $C_2H_6$ emissions between 2008 and 2014, and as a
result report a 3–5% annual increase in column $C_2H_6$ at Northern mid latitudes. They hypothesize that this increase
is the result of the recent massive growth in the exploitation of shale gas and tight oil reservoirs in North America,
where the drilling productivity began to grow rapidly after 2009.
**2. Methods**
**2.1 MkIV Instrument**
The JPL MkIV interferometer (Toon, 1991) is a high-resolution FITR spectrometer built at JPL in 1984. It covers
the entire 650–5650 cm⁻¹ range simultaneously in every spectrum with two detectors: a HgCdTe photoconductor
covering 650–1800 cm⁻¹ and an InSb photodiode covering 1800–5650 cm⁻¹. For ground-based observations a
maximum OPD of 117 cm is employed providing a spectral resolution of 0.005 cm⁻¹. The MkIV is primarily a
balloon instrument and has performed 25 flights since 1989, the latest in 2019. Between balloon flights it makes
ground-based observations. Since 1985 it has taken 5000 ground-based observations on 1200 different days from 12
different sites. For more detail, see tables in: https://mark4sun.jpl.nasa.gov/ground.html
**2.2 Retrieval**

Deleted: is

Deleted: report

Deleted: of essentially zero

Deleted: a

Deleted: .

Deleted: K

Deleted: K

The analysis of the MkIV spectra was performed with the GFIT (Gas Fitting) tool, a nonlinear, least-squares, spectral-fitting, algorithm developed at JPL.  GFIT has been previously used for the Version 3 analysis (Irion et al., 2002) of spectra measured by the Atmospheric Trace Molecule Occultation Spectrometer, and it is currently used for analysis of Total Carbon Column Observing Network (TCCON) spectra (Wunch et al., 2011) and for MkIV spectra (Toon et al., 2016; 2018a; 2018b). The entire package including spectral fitting software, spectroscopic linelists, and software to generate a priori VMR/T/P profiles, is termed GGG.

GFIT scales the atmospheric gas volume mixing ratio (VMR) profiles to fit calculated spectra to those measured. For $C_3H_8$, a 5.4 cm$^{-1}$-wide fitting window centered on the Q-branch at 2967 cm$^{-1}$ was used. The atmosphere was discretized into 70 layers of 1 km thickness. $C_3H_8$ and four interfering gases ($H_2O$, $CH_4$, $C_2H_6$, HDO) were adjusted. Two frequency stretches were retrieved (telluric and solar). The spectral continuum was fitted as a straight line, and a zero-level offset was fitted. So that's a total of 10 simultaneously-fitted scalars. In addition, the solar pseudo-transmittance was computed (but not adjusted).

The assumed temperature, pressure and $H_2O$ profiles were based on the NCEP 6-hourly analyses for solar noon of each day. The a priori vmr profiles were based on NH mid-latitude profiles. This is the same scheme as used by the GGG TCCON analysis (Wunch et al., 2015), but here we apply it to the Mid-IR MkIV spectra rather than the Short-Wave IR TCCON spectra.

To estimate the sensitivity of the retrieved $C_3H_8$ to uncertainties in the assumed a priori profiles of T/P and interfering gases (especially $H_2O$, $CH_4$), we retrieve the post-2000 $C_3H_8$ a second time: using GGG2020, an updated version of the GGG code with improved a priori VMR/T/P profiles based on the GEOS-FP-IT analysis (Laughner et al., 2021). The results, shown in figure B.2, illustrate that this changes the retrieved $C_3H_8$ by less than 10% with a bias of only 1.1%.

**2.3. Spectroscopy**

It is clear from the infra-red lab spectrum of $C_3H_8$ (Fig.1), measured at Pacific North-West National Laboratory (Sharpe et al., 2004), that the feature at 2967 cm$^{-1}$, caused by various $CH_2$ and $CH_3$ stretch vibrational modes, is by far the strongest in the entire infrared.  So for solar occultation spectrometry, this is by far the best choice. For thermal emission spectrometry from cold planets such as Titan, however, these bands are not covered by Cassini/CIRS since the thermal Plank function of such planets weakens rapidly above 2000 cm$^{-1}$. Thus, the much weaker bands below 1400 cm$^{-1}$ must be used (Sung et al., 2013).

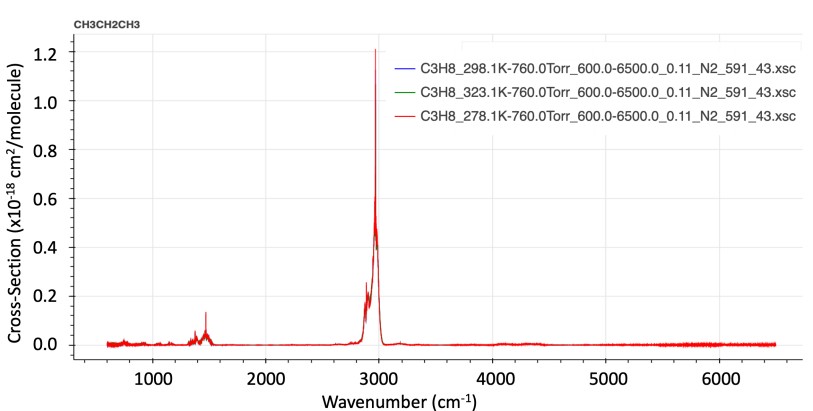

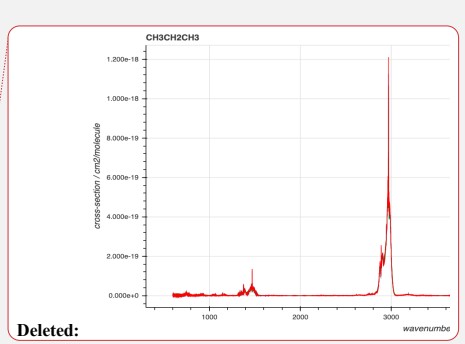

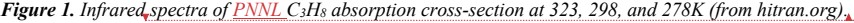


**Figure 1.** *Infrared spectra of PNNL C$_3$H$_8$ absorption cross-section at 323, 298, and 278K (from hitran.org).*

An empirical pseudo-line-list (EPLL) of C$_3$H$_8$ covering 2560–3280 cm$^{-1}$ was derived from the laboratory cross-
sections of Harrison and Bernath (2010b).  This is described in the unpublished report:
https://mark4sun.jpl.nasa.gov/data/spec/Pseudo/c3h8_pll_2560_3280.pdf
The use of an EPLL facilitates interpolation and extrapolation of the lab cross-sections to T/P conditions that were
not measured in the lab.  The fitting of the EPLL also checks the self-consistency of the lab cross-section spectra,
and provides an opportunity to correct for artifacts in the lab spectra (e.g., channeling, zero-level offsets,
contamination, ILS), although it must be stated that in this particular case the C$_3$H$_8$ lab spectra were of very high
quality and comprehensive in terms of their coverage.  For the interfering C$_2$H$_6$, an EPLL developed eight years ago
was used, based on lab measurements of Harrison et al. (2010a), as described in the report:
https://mark4sun.jpl.nasa.gov/report/C2H6_spectroscopy_evaluation_2850-3050_cm-1.compressed.pdf
For other gases the atm.161 linelist was used, which is based on HITRAN 2016, with some empirical adjustments
based on fits to lab spectra, especially for H$_2$O and CH$_4$.  This is basically the same linelists (atm.161, pll.101) that
are used by TCCON, but here we use them in the MIR rather than the SWIR.
Figure 2 shows an average spectral fit to the C$_3$H$_8$ window in ground-based MkIV spectra, obtained by fitting
individual spectra and then averaging the results. The lower panel provides the full transmittance y-range from 0 to
1. It can be seen that the main absorbers are CH$_4$ (orange) and H$_2$O (green).  The C$_3$H$_8$ absorption (red) is difficult to
discern because it is so shallow. The lower-middle panel shows the same spectral fit, but with the y-scale zoomed
into 0.95–1.00 transmittance, allowing the weak absorbers like C$_3$H$_8$ and C$_2$H$_6$ to be more easily seen. The "other"
contributions (e.g., O$_3$) were included in the calculation but not adjusted. The C$_3$H$_8$ absorption is fairly flat at about
1% depth, except for the Q-branch where it deepens to 2½%. Although the strongest C$_2$H$_6$ feature coincides with the
C$_3$H$_8$ Q-branch, the former is much narrower and there are several additional C$_2$H$_6$ features in this window, so the
spectrometric "cross-talk" between these two gases should be modest; we compute a Pearson Correlation
Coefficient of -0.7 between the $C_3H_8$ and $C_2H_6$. Further discussion on this topic can be found in Appendix A.  The
upper-middle panel shows that the residuals (measured-calculated transmittance) have some systematic features of
~0.5% in magnitude, especially in the vicinity of the $H_2O$ line at 2966.0 cm$^{-1}$.  The topmost panel shows the
residuals to a fit performed without any $C_3H_8$ absorption lines. It looks surprisingly similar to the fit performed with
$C_3H_8$ lines, such is the ingenuity of the spectral fitting algorithm in adjusting the $H_2O$, $CH_4$, and $C_2H_6$ to compensate
for the missing $C_3H_8$. The overall RMS residual in the no-$C_3H_8$ case is 0.3934%, as compared with 0.3658% when
$C_3H_8$ is included. This is quite significant considering that residuals are dominated by the $H_2O$ line at 2966.0 cm$^{-1}$,
and are unaffected by whether $C_3H_8$ is included or not. The residuals in the topmost panel (d) are larger in the
vicinity of the $C_3H_8$ Q-branch, 2967-2968 cm$^{-1}$, than those in panel (c).

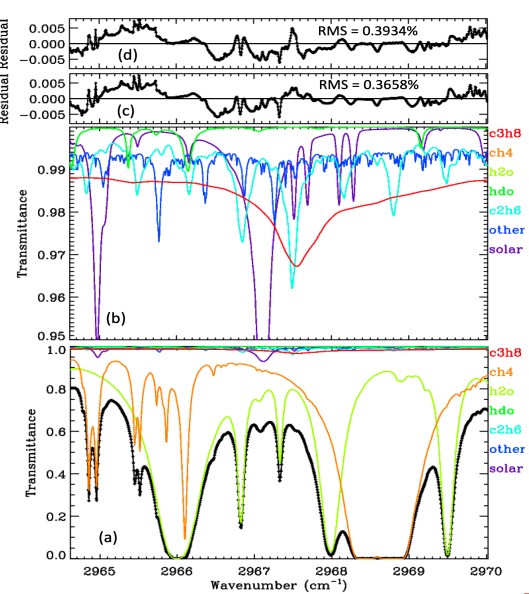

***Figure 2.***  *The average of 5000 ground-based MkIV spectral fits. Black diamonds represent measured spectrum.*
*Black line the fitted calculation. Colored lines represent the contributions of different gases. Panel (a) shows the full*
*transmittance range. Panel (b) zooms into the 0.95–1.00 range to help see the weak absorbers ($C_2H_6$, HDO, and the*
*solar lines). Panel (c) show residuals (Measured-Calculated); these are generally below 0.5%. Panel (d) shows the*
*residuals when $C_3H_8$ is excluded from the calculation.*
Considering the weakness (and smoothness) of the $C_3H_8$ Q-branch in comparison with the residuals and the
contributions of the other gases, we were at first skeptical that a useful $C_3H_8$ column measurement could be

extracted from such spectral fits.  But since the analysis of the MkIV spectra is highly automated, it took only a few hours to run the $C_3H_8$ window over all 5000 MkIV ground-based spectra.

**3. Results**

Table 1 lists the observation sites from where MkIV has made ground-based observations up to the end of 2019. The vast majority are from three sites: JPL, Mt. Barcroft, and Ft. Sumner.

| Town | State | $N_{obs}$ | $N_{day}$ | Latitude (deg.) | Longitude (deg.) | Altitude (km) | Terrain | Years Operated |
|---|---|---|---|---|---|---|---|---|
| Esrange | Sweden | 160 | 32 | 67.889 | +21.085 | 0.271 | Boreal | 1999–2007 |
| Fairbanks | Alaska | 124 | 46 | 64.830 | -147.614 | 0.182 | Boreal | 1997 |
| Lynn Lake | Manitoba | 20 | 11 | 56.858 | -101.066 | 0.354 | Boreal | 1996 |
| Mt. Barcroft | California | 1369 | 258 | 37.584 | -118.235 | 3.801 | Alpine | 1994–2002 |
| Mtn. View | California | 7 | 4 | 37.430 | -122.080 | 0.010 | Urban | 1987, 2001 |
| Daggett | California | 33 | 21 | 34.856 | -116.790 | 0.626 | Desert | 1993 |
| Ft. Sumner | New Mex. | 521 | 106 | 34.480 | -104.220 | 1.260 | Steppe | 1989–2019 |
| TMF | California | 475 | 45 | 34.382 | -117.678 | 2.257 | Alpine | 1986–2009 |
| JPL (B183) | California | 2273 | 690 | 34.199 | -118.174 | 0.345 | Urban | 1985–2020 |
| JPL (mesa) | California | 20 | 5 | 34.205 | -118.171 | 0.460 | Urban | 1988–1989 |
| Palestine | Texas | 4 | 3 | 31.780 | -95.700 | 0.100 | Rural | 1989 |
| McMurdo | Antarctica | 37 | 20 | -77.847 | +166.728 | 0.100 | Polar | 1986 |

*Table 1.* *The twelve sites from where MkIV has made ground-based observations, along with the number of observations and observation days from each site, years of operations, their location, and terrain type. They greyed out sites have the fewest observations (only 1% of total) and are not included in the Figures 5-7 and A.1 to reduced color ambiguity.*

Fig. 3 shows MkIV ground-based $C_3H_8$ columns, color coded by site altitude. The data were filtered: only points with uncertainties $< 1.5x10^{16}$ were plotted, reducing the number of plotted points from 5000 to 4700. The top panel (a) shows that at the high-altitude sites (Mt. Barcroft at 3.8 km is Red; Table Mountain Facility at 2.26 km is Orange) the retrieved $C_3H_8$ columns are centered around zero.  Also, the data acquired in Sep 1986 from 0.1 km in Antarctica (dark blue) are centered around zero.  Data acquired from Ft. Sumner, NM, at 1.2 km (lime) have large variations, from zero to nearly $8x10^{16}$ molecules.cm$^{-2}$, as do the data from JPL at 0.35 km (cyan). Other sites with detectable $C_3H_8$ include Daggett, CA, (0.6km), Esrange, Sweden (0.26km) in the winter, Fairbanks, AK (0.2km), and Mountain View, CA in late 1991. So $C_3H_8$ has only been measured by MkIV from northern hemisphere sites within the PBL. Panels (b) and (c) show the same $C_3H_8$ columns, but plotted versus year and day.

High $C_3H_8$ values ($>4x10^{16}$ molecules.cm$^{-2}$) can occur at any time of year at JPL (cyan) but most commonly in late summer, as is the case for other pollutants, e.g. CO. This reflects the meteorology (stagnant conditions in the LA

basin in summer with little replacement of polluted air with clean air from outside). Averaging kernels for these
$C_3H_8$ measurements are discussed and illustrated in Appendix B. Suffice it to say here that they range from 0.9 to
1.4 and increase with altitude.

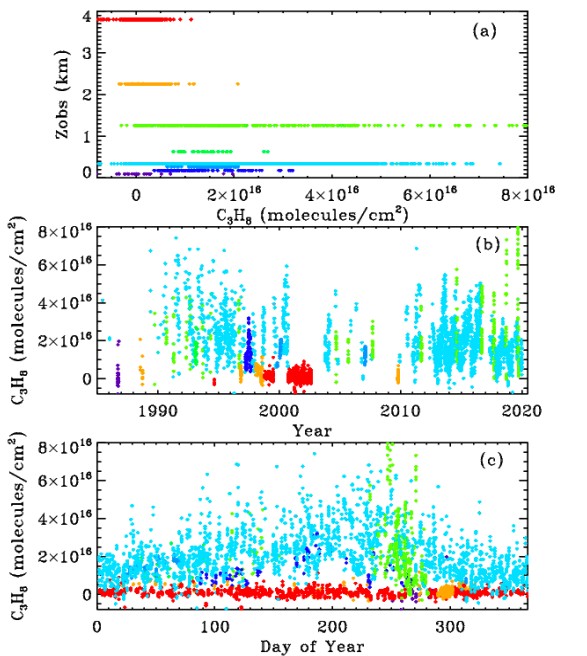

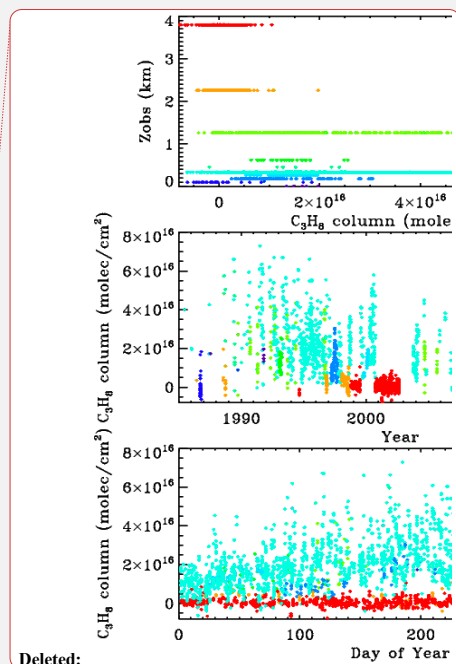


***Figure 3.*** *MkIV $C_3H_8$ column abundances from 8 out of 12 sites, color-coded by site altitude, as illustrated in panel*
*(a): Violet=0.1 km (McMurdo); dark blue=0.18 km (Fairbanks); light blue=0.27km (Esrange); cyan=0.35 km*
*(JPL); Green=0.63 km (Daggett); lime=1.2 km (Ft. Sumner); orange=2.26 km (TMF); red=3.8 km (Mt. Barcroft).*
The reported uncertainties in our $C_3H_8$ column measurements are based on the rms fitting residuals compared with
the sensitivity of the spectrum to $C_3H_8$ (Jacobians). At the highest site, Barcroft at 3.8 km (P=0.65 atm.), where the
interfering $H_2O$ and $CH_4$ absorptions are relatively weak and narrow, the $C_3H_8$ column uncertainties are generally
smaller than $10^{15}$ molecules.cm$^{-2}$. But since the columns themselves are even smaller, no $C_3H_8$ is detected at
Barcroft. At the lower altitude sites such as JPL and Ft. Sumner, the increased interference from $H_2O$ and $CH_4$ cause
the $C_3H_8$ column uncertainties to be much larger, generally around $5x10^{15}$ molecules.cm$^{-2}$ at low airmass and
worsening rapidly toward higher airmasses. But the $C_3H_8$ increases far more, allowing $C_3H_8$ to be detected at these
low-altitude sites under polluted conditions, despite the poorer absolute uncertainties.
High $C_3H_8$ values are also seen at Ft. Sumner, NM (lime), especially in recent years. This was initially a surprise to
us because this area has a very low population density, so we naively assumed that we would be measuring
background levels of atmospheric pollutants here.
We know that the apparent variations in $C_3H_8$ are real, rather than artifacts, from their strong correlation with $C_2H_6$.
Figure 4 compares column-averaged $C_3H_8$ mole fractions (top panels) with those of $C_2H_6$ (bottom panels), the latter
retrieved using different spectral lines than those shown in Fig.2. These are the same total $C_3H_8$ columns shown in
Fig.3, but divided by the total column of all gases, which is inferred from the surface pressure. The resulting
column-average mole fractions, denoted Xgas, are less sensitive to the site altitudes being different and more easily
compared with in situ measurements being in units of mole fraction.
The upper and lower rows of Fig.4 shows the XC3H8 and XC2H6 time series, respectively, plotted versus year (left)
and versus day of the year (right). The data were filtered such that only points with $XC_3H_8$ uncertainties < 0.74 ppb
and $C_2H_6$ uncertainties < 0.10 ppb were plotted. This reduced the total number of points from 5000 to 4700, so only
the best 94% of the data are plotted. It is clear that at JPL (cyan) $C_3H_8$ has decreased since the 1990s, but that at Ft.
Sumner (lime) it has increased over the past decade. The data from these two sites will be explored later.

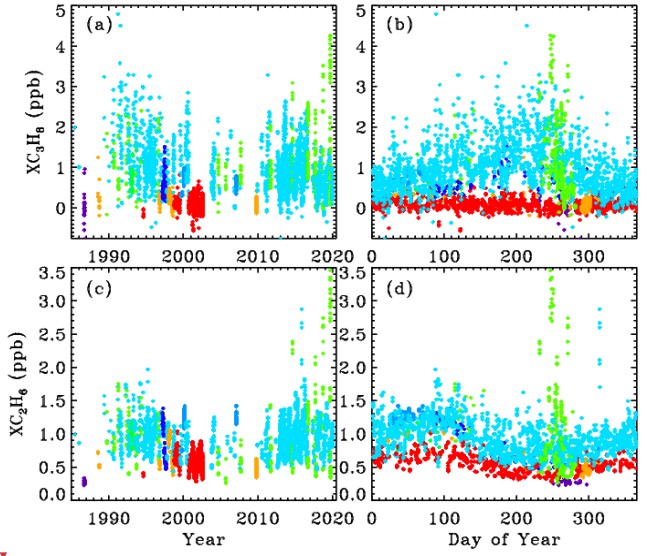


***Figure 4.*** *Top panels show measurements of the column-averaged $C_3H_8$ mole fractions ($XC_3H_8$). Bottom panels*
*show $XC_2H_6$. Left panels show the variation with year. Right-hand panels show the seasonal variation. Points are*
*color-coded by observation site altitude, as in Fig.3.*
$C_2H_6$ is four times longer-lived than $C_3H_8$ and never goes to zero because there is always a substantial free
tropospheric $C_2H_6$ component, even in the SH, which varies seasonally: high in spring, low in fall. The Antarctic
measurements (blue) are very low (0.2-0.3 ppb) and most probably even lower during the rest of the year, because
days 250 to 300 represent the springtime peak, not the fall. The highest $C_2H_6$ ever measured from JPL (cyan) was in
late 2015 (day 314) as a result of the Aliso Canyon natural gas leak (Conley et al., 2016). This event is further
discussed later and also in Appendix C.
Figure 5 shows the $XC_2H_6/C_3H_8$ correlation plot for all sites. This uses the exact same data, filtering, and color-
scheme as for Fig. 4.  At JPL (cyan) the correlation is positive but weak.  At Ft. Sumner, there are episodes of both
gases being enhanced with a strong correlation. In fact, the highest VMRs of $C_2H_6$ were seen from Ft. Sumner, even
more than from JPL during the Aliso Canyon gas leak in late 2015.

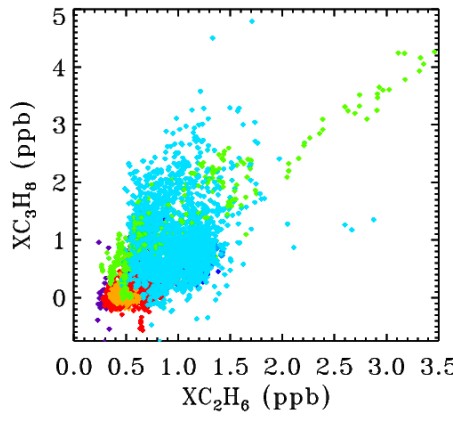

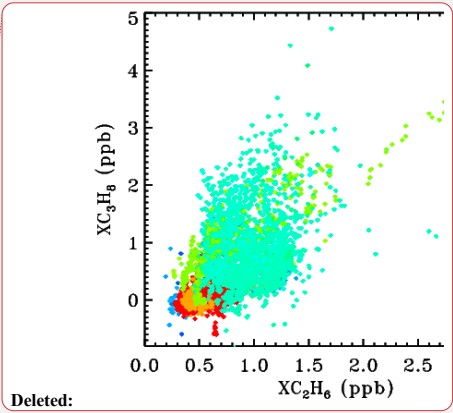


***Figure 5.*** *The correlation between $XC_2H_6$ and $XC_3H_8$ for all sites, color-coded by site altitude as in Fig.3.*
**3.1. Averaging Kernels.**
Figure 6 shows all kernels for the 5000 measurements presented in this paper, color-coded by site altitude (red=3.8
km; orange=2.2 km; lime=1.2 km; cyan=0.35 km; blue < 0.2 km) as in the main body of the paper. The kernels
increase with altitude but with <40% variation over the 0-30 km altitude range. Note that the kernels representing
the 3.8 km site begin at P=0.7 atm. And the kernels representing the 2.2 km site begin at P=0.8 atm.

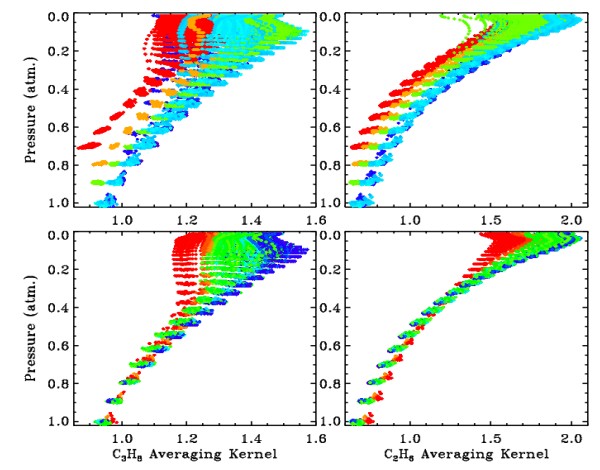

290

*Figure 6:* *5000 averaging kernels for **Left**: C₃H₈ and **Right**: C₂H₆. Upper panel shows all kernels color-coded by*
*site altitude, as in Fig.3. Lower panel shows kernels for the low-altitude sites (0.25 to 0.50 km), which were all*
*colored blue in the upper panel, now color-coded by solar zenith angle (Blue=15°; Green=60°; Red=80°).*

The lower panel shows the kernels for the low altitude sites (mainly JPL). These points were all cyan in the upper
panel but in the lower panel they are color-coded by Solar Zenith Angle. It is evident that the higher the SZA the
more uniform the kernels with altitude. The banding of the points in pressure space reflects the 1 km vertical grid on
which the kernels were computed. The $C_3H_8$ kernels are also influenced by the $H_2O$ column and temperature, but
these are smaller effects than those of site altitude or SZA.

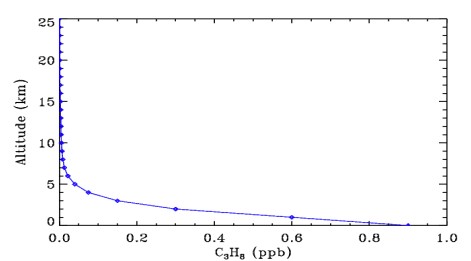

299

*Figure 7. A priori C₃H₈ profile used in these retrievals.*

Figure 7 show the assumed a priori vmr profile used in the retrievals and in the computation of the kernels. Since GFIT performs profile scaling retrievals, with a very weak a priori constraint, the absolute values of the vmrs play no role, only the profile shape matters.

**3.2 Case Study: Ground-based measurements from Ft. Sumner, NM**

Ft. Sumner (34.48N, 104.22W, 1.2 km ASL) is the location of the main NASA facility for the launch of stratospheric research balloons. It is located here due to the low population density and hence low risk of mishap. The MkIV instrument has performed balloon campaigns in Ft. Sumner 18 times in the past 30 years. Not all of these campaigns have resulted in a flight, but we have always taken ground-based observations to check that the MkIV instrument is correctly aligned and functional, and to check that telemetry, commanding, and the operation of other experiments do not degrade the MkIV performance.

We have taken 520 observations on 106 different days from Ft. Sumner (out of a total of 5000 observations and 1200 days). We examine these observations to try to understand whether the large day-to-day $C_3H_8$ variations are real, and if so, what is causing them. We have already seen a correlation between the $XC_3H_8$ and $XC_2H_6$ at all sites in Fig.5, but many points are buried under others, especially at the low values of $XC_3H_8$ and $XC_2H_6$.

*__Figure 8__ XC$_3$H$_8$, XC$_2$H$_6$ and XCO at Ft. Sumner. Since all the observations are made from the same altitude, it no longer makes sense to color code by site altitude. So instead we color-code by mean bearing of the back-trajectory over the previous 36 hours. Dark blue=30°; Light blue =90°; Cyan=120°; Green=180°; Lime=220°; Orange= 300°; Red=350°. __MKIV didn't visit Ft. Sumner from 1997 to 2004 because it was performing high-latitude balloon flights from Alaska and Sweden.__*

Figure 8 shows that between 1990 and 2005 there was a decrease in C$_2$H$_6$ and C$_3$H$_8$ measured in Ft. Sumner, by about a factor 2 over 15 years. In recent years (since 2014), however, there has been a large increase in C$_2$H$_6$ and C$_3$H$_8$ measured at Ft. Sumner, but only when the wind direction is from the SE quadrant (green-cyan colors). We see no increase associated with other wind directions (red, blue, orange, yellow, lime).

At Ft. Sumner CO has no correlation with wind direction, nor with C$_2$H$_6$ or C$_3$H$_8$. The majority of days have a column average CO of 75±10 ppb. But there are occasional enhancements up to 120 ppb, likely due to large but distant fires. We do not pursue the Ft. Sumner CO data any further, beyond proving that the C$_3$H$_8$ sources are different from those of CO.

CH$_4$ is also measured by MkIV. Over the 30-year measurement period XCH$_4$ has grown from 1650 to 1850 ppb. This secular increase is much larger than any variation due to wind direction. So to be useful, the CH$_4$ data would have to be detrended, which is not simple given its non-linear growth. Even within the past 4 years, the correlation of XCH$_4$ with XC$_3$H$_8$ was very weak. This is to be expected since the background abundance of CH$_4$ is more than 1000x larger than C$_3$H$_8$, whereas wet NG is only 6 times richer in CH$_4$ than C$_3$H$_8$ (in the Permian basin). So the NG-induced enhancement of CH$_4$, as a fraction of its atmospheric background level, will be much smaller than that of C$_3$H$_8$.

Figure 9 shows a XC$_3$H$_8$-XC$_2$H$_6$ scatter plot using just the Ft. Sumner data. Error bars are much larger for XC$_3$H$_8$ than for XC$_2$H$_6$. This is because the C$_2$H$_6$ transitions are stronger and form narrower features, both of which make the retrievals more precise and definitive, whereas most of the C$_3$H$_8$ absorption is smeared into a broad continuum which provides little information for a retrieval in which the continuum level is fitted. The C$_2$H$_6$ features used in the actual C$_2$H$_6$ retrieval are at 2976.6 and 2986.6 cm$^{-1}$ (not shown) and are 3–4 times stronger than those seen in Fig.2.

**Moved down [5]:** *MKIV didn't visit Ft. Sumner from 1997 to 2004 because it was performing high-latitude balloon flights from Alaska and Sweden.*

**Deleted:** *6*

**Moved (insertion) [5]**

**Deleted:** ,

**Deleted:** 6

**Deleted:** /lime

**Deleted:** . They are of no value

**Deleted:** other than

**Deleted:** from the Permian Basin

**Deleted:** .

**Deleted:** be more than 100 times

**Deleted:** 7

**Deleted:** this type of

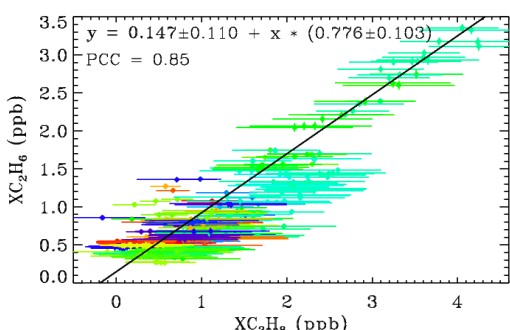

360

**Figure 9.** *The relationship between $XC_3H_8$ and $XC_2H_6$ at Ft. Sumner, color coded for wind direction as for Fig.8.*

The gradient of the fitted line is 0.78±0.10 implying more $C_3H_8$ than $C_2H_6$. The Pearson Correlation coefficient is 0.85, which is high considering the large error bars on the $XC_3H_8$ and the fact that the similarity of their Jacobians would imply an anti-correlation in their retrieved amounts (Appendix A).  This tight relationship at Ft. Sumner suggests that the large variations in the $C_3H_8$ measurements are not an artifact. Since $C_2H_6$ can be easily and precisely measured by this technique, it is hard to imagine it being changed by a factor 5 from day to day by an artifact.  Much more likely, the common variations in both $C_3H_8$ and $C_2H_6$ are real.

As already hinted, for each of the 106 observation days from Ft. Sumner we ran hourly HYSPLIT back-trajectories (Stein et al., 2015, Rolph et al., 2017) that bracket the MkIV observation times, then interpolated linearly in time between the two bracketing trajectories.  This provided a unique trajectory for each of the 520 observations from Ft. Sumner.  The North American Regional Reanalysis (NARR) meteorology was selected which covers North America at 32 km resolution. This is the highest resolution meteorology that covers the entire 1989–2019 observation period. A trajectory altitude of 0.4 km over Ft. Sumner was selected, and these trajectories were extended to 36 hours before the observations in 1-hour steps.  Fig.10 shows that the large variations of $C_3H_8$ are strongly correlated with wind direction. It is very clear that trajectories originating to the SE of Ft. Sumner, carry more $C_3H_8$ than those from any other direction. A plot was made also for $C_2H_6$ but not shown due to its strong similarity to Fig.10.

Deleted: 7
Deleted: 6
Deleted: .
Deleted: implie
Deleted: 8
Deleted: 8

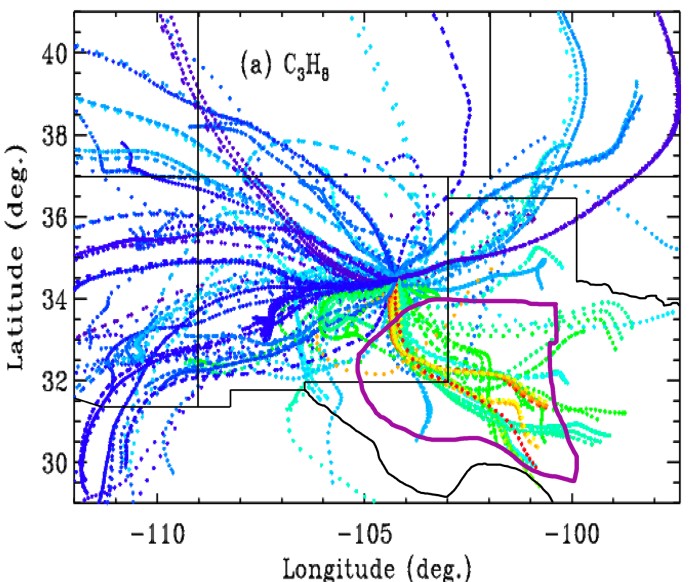

383

**Figure 10.** *Hourly locations for the back-trajectories, color-coded by retrieved XC₃H₈. Blue=0 ppb, Green=2 ppb;*
*Red=4 ppb. Trajectories for which the XC₃H₈ uncertainty exceeded 0.74 ppb are excluded, resulting in only 373 out*
*of 520 trajectories being shown. Ft. Sumner lies at 34.2N, 104.2W, close to the center of the figure at the confluence*
*of all the back-trajectories. Each point represents a 1-hour time step, so that the wind speed is apparent from the*
*separation of points. Winds from the West are typically stronger than those from the SE quadrant. Trajectories are*
*underlaid by a map of New Mexico and neighboring states. The Permian Basin, encircled by the thick purple line,*
*underlies SE New Mexico and much of West Texas. Many of the trajectories from the SE have spent 30+ hours over*
*the Permian Basin.*

We also made a scatter plot for CO (not shown) but there was no correlation between CO and wind direction, or
between CO and $C_3H_8$. This rules out the possibility that the enhanced $C_3H_8$ and $C_2H_6$ were somehow associated
with distant urban pollution or wild fires.

This result leads to speculation on what might be enhancing $C_2H_6$ and $C_3H_8$ when the winds come from the SE
sector. One of the biggest natural gas production fields in the US lies in the Permian Basin, which underlies the
South-East corner of New Mexico and West Texas, as illustrated in Figure 10. This region also includes processing
plants where the heavier gases are stripped out of the wet NG, storage facilities for the resulting Natural Gas Liquids
(LPG+ethane+pentane), and pipelines. The Permian Basin is by far the largest "liquids-rich" (rich in heavy
hydrocarbons) gas field in the USA (https://www.spglobal.com/platts/plattscontent/_assets/_images/latest-

news/20191219-rig-count.jpg). This would suggest that the enhanced $C_2H_6$ and $C_3H_8$ is the result of losses from NG
production, although this cannot be proven with just one instrument at one site. We would need instruments upwind
and downwind to make an accurate assessment of the fluxes.

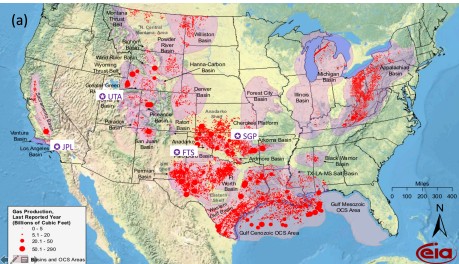



**Figure 11**: *NG production in the lower 48 states of the USA in 2009. Data from the Energy Information*
*Administration: https://www.eia.gov/oil_gas/rpd/conventional_gas.pdf. Superimposed are the locations (purple*
*pentangular star) of the four sites discussed in detail in this paper: Ft. Sumner in Eastern NM is labelled "FTS". The*
*JPL site in California is labelled "JPL". The locations of the NOAA sites in Utah (UTA) and Oklahoma (SGP) are*
*also included. The Permian basin lies in the SE corner of NM and West Texas.*
The Permian basin currently produces 16 billion cu.ft./day of NG
(https://www.eia.gov/petroleum/drilling/pdf/permian.pdf) over an area of 220,000 km². The molar volume of an
ideal gas at STP is 22.4 liters. One cu. ft. is 28.3 liters. So 16 billion cu. ft. is 20 billion moles of NG or $120x10^{32}$
molecules per day.  Over an area of 220,000 km² or $2.2x10^{15}$ cm², this represents an average areal production of
$55x10^{17}$ molec./cm²/day. Assuming that the Permian basin is 480 km wide, at an average low-level wind speed of 15
km/hour, an air parcel will take 32 hours (1.33 days) to traverse the Basin, during which time $73x10^{17}$
molecules/cm² will have been extracted. Of this, 10% will be $C_3H_8$ (Howard et al., 2015), so if all this production
were released into the atmosphere we would expect a $C_3H_8$ column enhancement of $73x10^{16}$.
In airmasses with trajectories from the SE, we see maximum $C_3H_8$ column enhancements of only $3x10^{16}$
molecules/cm², which suggests that only 4% of the NG escapes into the atmosphere and that 96% of the NG is
successfully captured (or burnt by flaring).
In the Permian Basin, NG is 13.7% $C_2H_6$ and yet the observed ethane enhancements are slightly smaller than those
of $C_3H_8$, suggesting that only ~3% of the NG escapes. Assuming a 3% leak rate, there will also be an enhancement
of $CH_4$ of about $14x10^{16}$ molec.cm⁻², but this represents only 0.4% of the total $CH_4$ column above Ft. Sumner and
will therefore be difficult to discern in the presence of other confounding factors (stratospheric transport, varying
tropopause altitude, seasonal and longer-term changes).  Of course, all this analysis assumes that the Permian basin

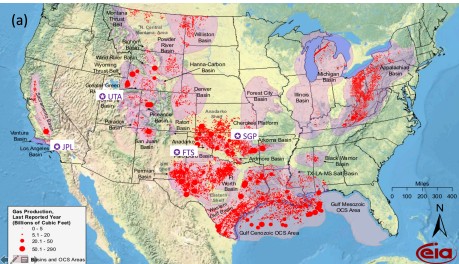

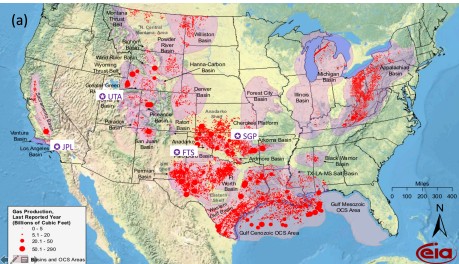

Deleted: *9*

Deleted: *(a)*

Deleted: ¶
 *(b): [Temporarily removed -- awaiting permission] Illustrating the high number of "liquids-rich" drilling rigs in the Permian Basin, as of Dec. 2019, underscoring its dominance for propane production in the USA. From https://www.spglobal.com/platts/en/market-insights/latest-news/natural-gas/121919-us-oil-gas-rig-count-rises-for-second-straight-week-enverus¶*

Formatted: Font: Times New Roman, 10 pt

Formatted: Line spacing:  1.5 lines

Formatted: Font: Times New Roman, 10 pt

Formatted: Font: Times New Roman, 10 pt

Formatted: Font: 10 pt

Deleted: In recent years, the Permian basin has been producing ~15 billion cu.ft. of natural gas (NG) per day (https://www.eia.gov/petroleum/drilling/pdf/permian.pdf). A back-of the-envelope estimate of the contribution of this to the observed $C_3H_8$ is now performed. We assume that this NG production is distributed over an area that is 160 km across. At a wind speed of 20 km/hour, an airmass will take 8 hours to traverse the gas field, during which time 0.72E+19 molecules.cm⁻² of NG will have been extracted.  Howard et al., (2015), measured the composition of NG from the Permian basin and found that it is very rich in heavy hydrocarbons, being 66.6% $CH_4$, 13.7% $C_2H_6$ and 10.3% $C_3H_8$ by volume. If 4% of this were lost to the atmosphere, and 10.3% of this is $C_3H_8$, the total propane column will be enhanced by 3E+16 molecules.cm⁻², which is close to that seen in the highest cases. For

Deleted: $C_2H_6$,

Formatted: Subscript

Formatted: Subscript

Deleted: an

Deleted: of 4E+16

Deleted:

Deleted: would be expected for such a back-trajectory, which is somewhat higher than measured

Deleted: T

Deleted: 9E+

Deleted: 5

is a uniform emitter and that the back trajectory wind speeds are accurate. There are likely hot spots with higher-than-average emissions, and regions with little NG production.

A puzzle in our findings is that when both $C_3H_8$ and $C_2H_6$ are elevated, we measure 22% more $C_3H_8$ than $C_2H_6$ (see fig.9). Yet independent essays of well-head wet NG find 33% more $C_2H_6$ than $C_3H_8$ in the Permian basin (Howard et al., 2015). So we have a 55% discrepancy. We note that the $C_2H_6$ averaging kernel is 0.7 at the surface versus 0.9 for $C_3H_8$ (see Appendix B). So when these gases exceed their priors in the PBL, which is likely at high enhancements, both will be under-estimated, but $C_2H_6$ more so than $C_3H_8$. So this effect would cause the $C_3H_8$/$C_2H_6$ ratio to be 28% high, which explains half the 55% problem. Another possibility is that the $C_3H_8$ coming from fugitive wet NG is augmented by leaks of LPG, stripped from wet NG. This would further enhance the $C_3H_8$ (and $C_4H_{10}$) with little $C_2H_6$ increase. Alternatively, there could be a systematic over-estimate of the MkIV $C_3H_8$ due to a mundane multiplicative bias in the $C_3H_8$ spectroscopy. This would over-estimate all the $C_3H_8$ measurements without degrading the strong correlation with $C_2H_6$, but seems unlikely.

**3.3. Case Study: Ground-based measurements from JPL**

The Jet Propulsion Laboratory (34.2N; 118.17W; 0.35 km altitude) lies at the Northern edge of the Los Angeles basin. When winds are from the North (rare in summer) air quality is good. When conditions are stagnant (common in summer) pollutants accumulate and so air quality is poor. $C_3H_8$ measured at JPL exhibits very different behavior to that at Ft. Sumner. It decreases over time, exhibits little correlation with $C_2H_6$, and positive correlation with CO. Figure 12 illustrates these behaviors.

The left-hand panels of Fig. 12 shows $XC_3H_8$ time series measured from JPL, color coded by CO. The upper-left panel shows a large decrease in $C_3H_8$ from 1–3 ppb in 1990 to less than 1 ppb in 2019. This mirrors the decrease in CO over JPL (not shown) over the same period. The lower-left panel shows a large seasonal component to the $C_3H_8$, with a peak in late summer, when the air is most stagnant over JPL allowing pollutants to accumulate. The highest $C_3H_8$ values appear red or orange (high CO), while the lowest appear blue (low CO), implying an association with CO. This is confirmed in the upper-right panel which plots $C_3H_8$ directly against CO. The right-hand panels are color-coded by year. The $C_3H_8$ correlation is mostly a result of both gases having decreased over the 30-year record. But even within each year, there still remains a positive correlation. This does not necessarily mean that $C_3H_8$ and CO have the same source, but that their sources are spatially coincident.

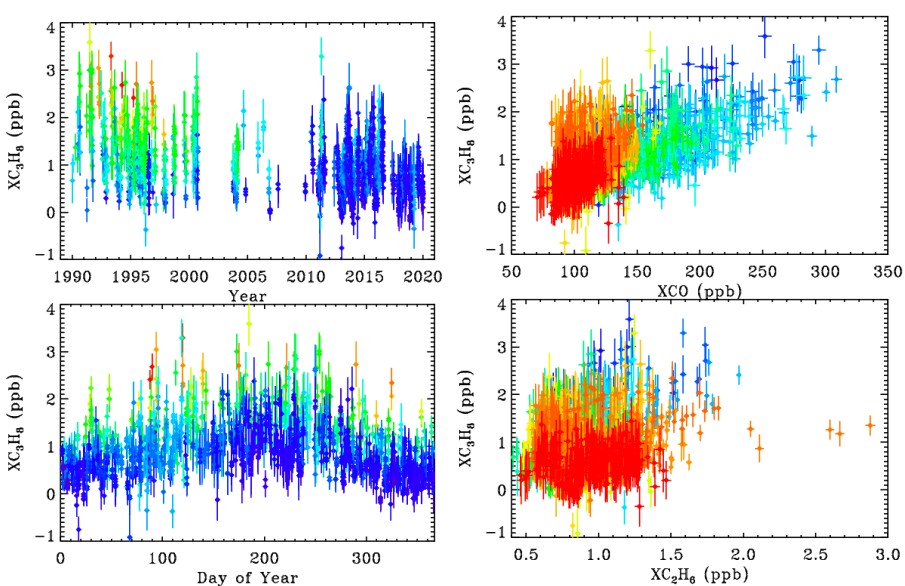

519

*Figure 12. Column-average C₃H₈ above JPL. **Left Panels:** The time series color-coded by CO (red=250 ppb;*
*green= 130 ppb; blue=100 ppb). **Right Panels:** The relationship between XC₃H₈ and CO and C₂H₆ color-coded by*
*year (blue=1990; green=2005; red=2019).*

The lower-right panel shows $C_3H_8$ plotted versus $C_2H_6$. There is a weak correlation at JPL. The high $XC_2H_6$ values
exceeding 2.0 ppb were measured in day 314 of 2015 when JPL was downwind of the Aliso Canyon NG leak.
Appendix C shows a HYSPLIT back-trajectory confirming this assertion. This spike can also be seen in Fig. 3.
There is no $C_3H_8$ enhancement associated with the $C_2H_6$ spike, since processed NG was leaking from an
underground storage facility, the heavy hydrocarbons (e.g., $C_3H_8$, $C_4H_{10}$) having already been stripped out. A 2%
increase in column-averaged $CH_4$ was also noted in the plume of the Aliso Canyon leak, as shown in Appendix C.

California accounts for less than 1% of total U.S. natural gas production and this has declined over the past three
decades (https://www.eia.gov/state/analysis.php?sid=CA). Although there is natural gas extraction in the LA basin,
this is a small source compared with the Permian basin. The local natural gas is only 3% $C_2H_6$ and 0.3% $C_3H_8$,
(https://www.socalgas.com/stay-safe/pipeline-and-storage-safety/playa-del-rey-storage-operations) and so cannot
account for the approximately equal amounts of these gases measured at JPL by the MkIV. We speculate that the
$C_3H_8$ measured at JPL comes mainly from LPG (e.g., used in "clean" commercial vehicles, BBQ grills, external
heaters, etc.). We can certainly rule out the possibility that the $C_3H_8$ measured at JPL is the result of wild fires, since
these have increased in recent years whereas the $C_3H_8$ has decreased.

### 3.4. Comparison with In Situ Measurements

First it should be pointed out that the column-average mole fractions that are derived from the column measurements will under-estimate the gas amount in the PBL for gases like $C_2H_6$ and $C_3H_8$ that reside mainly in the PBL. For example, if $C_3H_8$ resides entirely between 1000 and 800 mbar, with none in the free troposphere or stratosphere, then the column-average values will be 5 times smaller than the actual mole fractions in the PBL. So direct comparisons of the remote and in situ mole fractions should be avoided. But their behavior as a function of year or season, or gas-to-gas correlations, can still be meaningfully compared. This effect is in addition to the effect of their averaging kernels being less than 1.0 at the surface, which was discussed earlier.

In situ $C_3H_8$ and $C_2H_6$ mole fractions from the Wendover, Utah (UTA) and Southern Great Plains, Oklahoma (SGP) sites were downloaded from the NOAA Global Monitoring Laboratory website: (https://www.esrl.noaa.gov/gmd/dv/data/). These sites are the closest to Ft. Sumner. These are surface flask measurements covering the period 2006 to 2017. Figure 13 illustrates these data as a function of the year (left panels), the day of the year (middle panels), and the $C_3H_8$–$C_2H_6$ relationship (right panels). The upper panels cover the UTA site and the lower panels the SGP site. Note the factor 10 change in the y-scale: there is 10x more of these gases at SGP than at UTA. Looking at the map in Fig. 11, this is clearly because SGP lies immediately downwind of the Anadarko Basin oil and NG fields under the prevailing WSW winds. In contrast, the UTA site has no major up-wind source.

These in situ measurements confirm that $C_3H_8$ is highly variable with large enhancements being associated with oil and NG production fields. At SGP the $C_3H_8/C_2H_6$ ratio is about 0.65. This is smaller than those measured by the MkIV, but NG in the Permian basin is much wetter (richer in $C_3H_8$) than in the Anadarko basin.

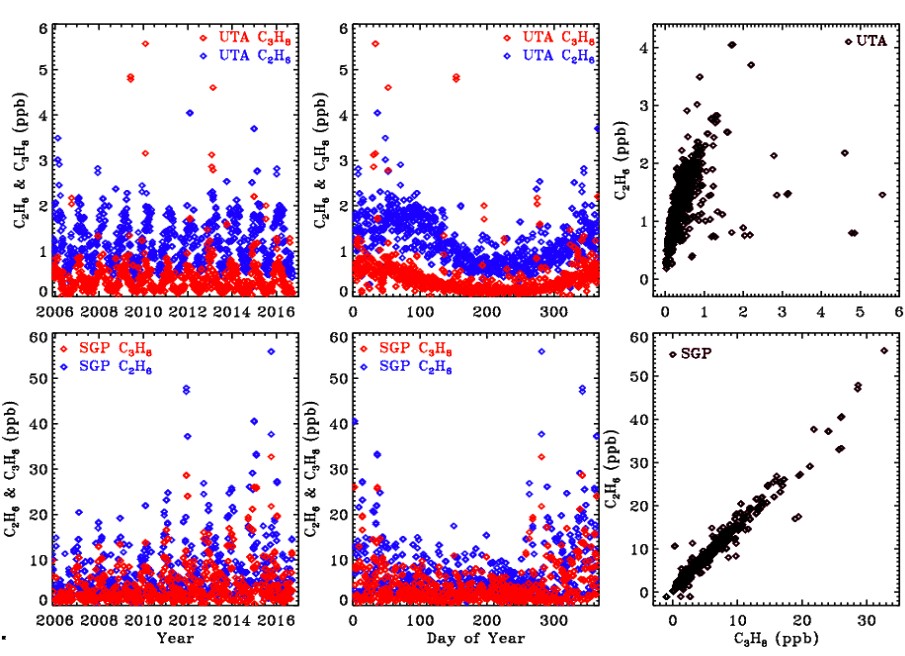

**Figure 13.** *In situ flask measurements of C₃H₈ (red) and C₂H₆ (blue) from the NOAA ESRL GMD dataset (Helmig et al., 2017). Top panels show results from the UTA site and lower panels from SGP.  Note the factor 10 change in the y-scale between the two sites. Left panels plot data versus year to illustrate secular trends. Middle panels versus Day of year to more clearly see the seasonal cycle. Right panels plot C₃H₈ versus C₂H₆.*

### 3.5. Balloon Results

We also attempted to retrieve $C_3H_8$ from MkIV balloon solar occultation spectra. It was not detected in any flight, despite a very good sensitivity of 0.05 ppb above 5 km. This confirms that the $C_3H_8$ detected in ground-based measurements, reaching column average mole fractions of up to 4 ppb, resides mostly in the PBL.  The balloon launches are typically performed only under stable, quiescent, meteorological conditions with light surface winds. Such conditions preclude uplift of air from the PBL into the free troposphere, so that $C_3H_8$ stays confined to the PBL, which is opaque in limb paths due to aerosol, and so cannot be probed in occultation. This does not preclude $C_3H_8$ getting up into the free troposphere at other times or in other places.

**4. Summary and Conclusions**

We report measurements of atmospheric $C_3H_8$ by solar absorption spectrometry in the strong Q-branch region at 2957 cm$^{-1}$, using high resolution IR spectra from the JPL MkIV interferometer. To the best of our knowledge, these are the first remote sensing measurements of atmospheric $C_3H_8$. The minimum detectable abundance is about $10^{16}$ molecules.cm$^{-2}$, which is roughly equivalent to a column average mole fraction of 0.5 ppb. This allows $C_3H_8$ to be measured in locations where its abundance is enhanced by proximity to sources (e.g., large gas fields, mega-cities), but not in clean locations (e.g. above the PBL or away from sources). We encourage such NDACC and TCCON sites to examine their datasets for $C_3H_8$. Future improvements to the spectroscopy of the interfering gases, e.g. $H_2O$, $CH_4$, $C_2H_6$, and other CH-containing gases currently missing might even provide for the detection of $C_3H_8$ from clean sites at background levels, allowing it to become a routine product of the NDACC and TCCON networks.

A case study of ground-based MkIV measurements from Ft. Sumner, New Mexico, shows increasing $C_3H_8$ and $C_2H_6$ amounts in the past decade on days when back-trajectories came from SE New Mexico and West Texas, where the Permian Basin oil and gas field is located. A case study of $C_3H_8$ measured at JPL shows a long-term decrease since 1990 by more than a factor 2. It also shows a strong correlation with CO, a tracer of urban pollution. There is no significant correlation between $C_3H_8$ and $C_2H_6$ at JPL.

The MKIV measurements in the case studies are not particularly useful for determining the long-term global trends in $C_3H_8$ or $C_2H_6$, due to their close proximity to strong sources. In the case of the Ft. Sumner the source is the Permian Basin. In the case of JPL the source is the Los Angeles urban area with a population of ~15M. These sources cause large meteorology-driven fluctuations that mask the longer-term trends.

From balloon measurements in solar occultation, propane was analyzed using the same window as for the ground-based measurements. It was not detected at any altitude in any of our 25 flights, despite a 0.05 ppb detection limit. This is presumably because under the stable atmospheric conditions that allow balloon launches, $C_3H_8$ stays confined to the PBL, which is opaque in the limb viewing geometry and so cannot be probed.

**Appendix A: Correlations between retrieved parameters**

We compute Pearson Correlation Coefficients (PCC) from the a posteriori covariance matrix for each of the 5000 spectral fits. The figures on the upper-left show the PCC between retrieved $C_3H_8$ and $C_2H_6$. Points are plotted versus year with the same site-altitude-dependent coloring as in the other figures. The PCC between $C_3H_8$ and $C_2H_6$ averages about -0.7, which means that they are fairly strongly anti-correlated. This is due to their overlapping absorption features at 2967.5 cm$^{-1}$. So as retrieved $C_2H_6$ increases, retrieved $C_3H_8$ will decrease, and vice versa. The PCCs are closer to zero for the high-altitude sites (red & orange), presumably due to the reduced pressure broadening and $H_2O$ causing the $C_2H_2$ and $C_2H_6$ absorption features to become more distinct. This anti-correlation could be reduced by use of a wider window to introduce additional $C_2H_6$ features that don't correlate with $C_3H_8$, but this would also encompass large residuals without adding any $C_3H_8$ information.

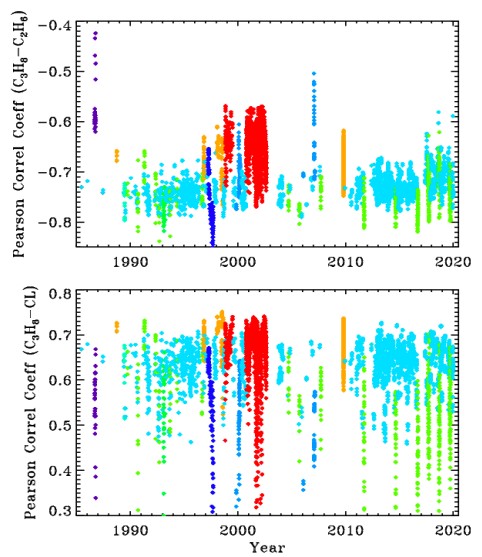

**Figure A.1**, *Pearson Correlation Coefficients between $C_3H_8$ and $C_2H_6$ (upper panel) and between $C_3H_8$ and the continuum level (CL)(lower panel)*

The $C_3H_8$-CL correlations are about +0.65 at low SZA, decreasing at higher SZA as the $H_2O$ and $CH_4$ absorptions black out the window. So the more $C_3H_8$ that is retrieved, the higher the continuum level has to be to match the measured spectrum, due to the fact that the $C_3H_8$ absorption spectrum has a broad continuum-like component beneath the Q-branch. The PCCs between $C_3H_8$ and the other retrieved parameters (e.g. $H_2O$, HDO, $CH_4$, Continuum Tilt, Frequency Shifts) were all much closer to zero than with $C_2H_6$ and CL.

The high PCC between $C_3H_8$ and $C_2H_6$ doesn't necessarily imply a large uncertainty in the $C_3H_8$. It just means that the large component of the $C_2H_6$ uncertainty gets projected onto the $C_3H_8$. Ditto for the CL. But provided the $C_2H_6$ and CL are well retrieved, their effect on the $C_3H_8$ will not dominate.

**Appendix B: Sensitivity of retrieved $C_3H_8$ columns to assumed P, T, and $H_2O$ profiles.**

The retrievals shown in the main body of the paper were performed using 6-hourly NCEP analyses of T, P, and $H_2O$, as used in the GGG TCCON analyses (Wunch et al., 2011). Due to the overlap of strong $H_2O$ and $CH_4$ lines with the $C_3H_8$ Q-branch, we were concerned that small errors in the assumed T/P/$H_2O$/$CH_4$ priors might strongly influence the retrieved $C_3H_8$. We therefore re-retrieved $C_3H_8$ over the 2000–2020 period using the GEOS-FP-IT 3-hourly

analyses, which forms the basis of the latest (GGG2020) TCCON analysis (Laughner et al., 2021). We would have done the entire analysis with the GEOS-FP-IT model, except that it only supports the post-2000 time period.

Figure B.1 compares the retrieved $C_3H_8$ columns from the two analysis methods: NCEP in the left panels and GEOS-FP-IT in the right-hand panels. The results look very similar.

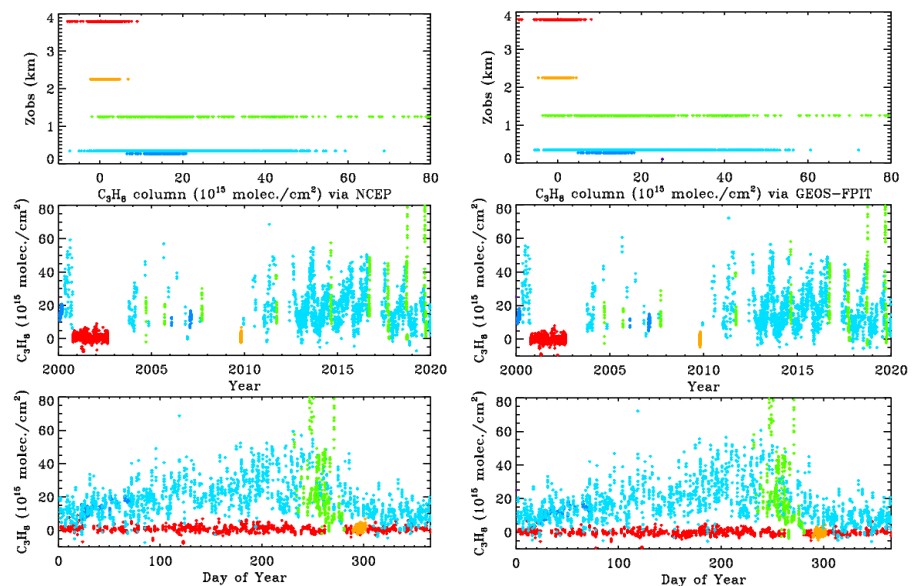

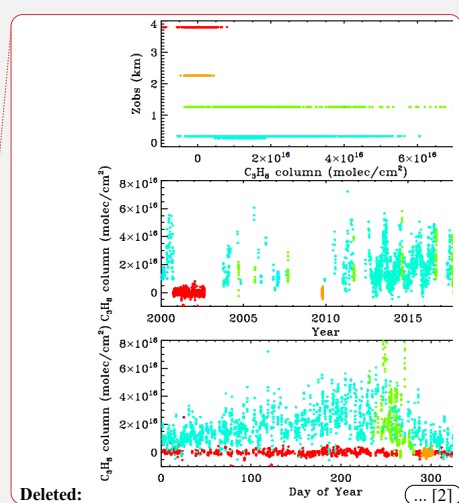

*Figure B.1.* *Retrieved vertical columns of $C_3H_8$ from 2000 to 2020 using two different atmospheric models.* ***Left:*** *NCEP a priori T/P/H_2O.* ***Right:*** *GEOS-FPIT a priori T/P/H_2O. Points are color-coded by site altitude, as in Fig.3.*

Figure B.2 examines more closely the $C_3H_8$ columns from the two analyses. In the upper panel the NCEP and GEOS-FPIT columns are plotted against each other. The gradient is 1.011±0.003 with NCEP producing slightly larger columns. The Pearson correlation coefficient is +0.979. The column differences, shown in the lower panel, are mostly less than $5\times10^{15}$ and are centered around zero at all column amounts. So the choice of models and priors makes surprisingly little difference to the retrieved $C_3H_8$. This does not mean that the $C_3H_8$ is highly accurate. There are many things that are identical between the two analysis (e.g., spectroscopy, retrieval code, spectra) which could

nevertheless contribute large errors to the retrieved C$_3$H$_8$.

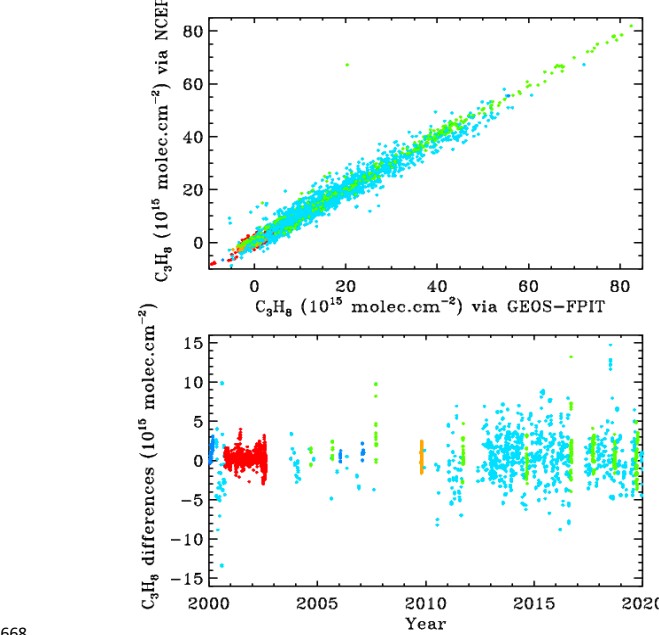


***Figure B.2.*** *Comparing the C$_3$H$_8$ columns retrieved from the 6-hourly NCEP and the 3-hourly GEOS-FPIT priors,*
*color-coded by site altitude. In the upper panel the columns are plotted against each other. In the lower panel their*
*difference is plotted.*
**Appendix C: - Aliso Canyon Underground Storage Facility: Gas Leak in late 2015**
Aliso Canyon Underground Storage Facility is located 30 km NW of JPL. According to the Jan 4, 2016,
Los Angeles Times, NG leak began Oct 23, 2015 and peaked on Nov 28 at 60 Tons of CH$_4$ per hour. By Dec 22 leak
rate had decreased to 30 Tons per hour as the underground storage pressure dropped from the initial 2700 psi.

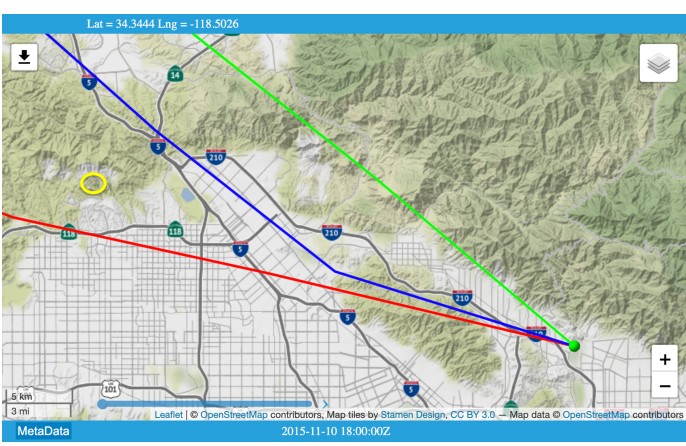

714

***Figure C1.** HYSPLIT back-trajectories for Nov 10, 2015 (day 314) when the highest ever $C_2H_6$ was measured from JPL. Yellow oval (upper-left) indicates location of Aliso Canyon Underground Storage Facility. Green ball (lower-right) denotes JPL, at the convergence of trajectories arriving at 19, 20, & 21 UT. Trajectory calculation used the NAM 12 km resolution, hybrid sigma-pressure meteorology. © OpenStreetMap contributors 2020. Distributed under a Creative Commons BY-SA License.*

Large $C_2H_6$ amounts (3x normal) were observed from JPL on Nov 10 (Day 314), but no enhancement of $C_3H_8$. HYSPLIT back-trajectories for this day indicate that the air arriving at JPL at 1000m above ground was from the North-West and had passed over Aliso Canyon USF, confirming that the air over JPL was contaminated by the leak.

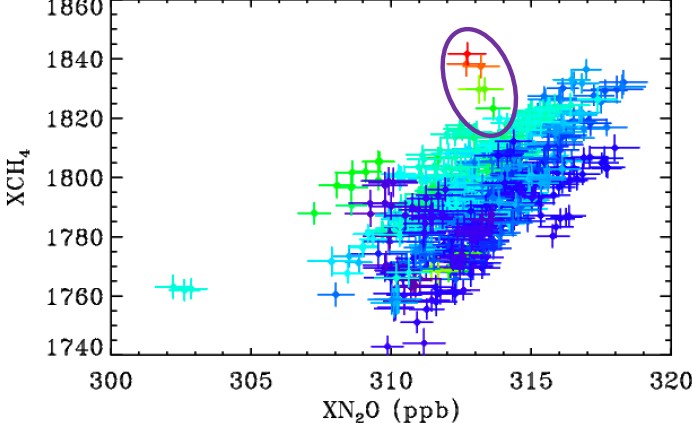

724

*Figure C.2. Showing the relationship between $CH_4$ and $N_2O$ at JPL in 2014–2017 color-coded by $C_2H_6$. Blue points represent low $C_2H_6$ whereas red represents the highest $C_2H_6$. The encircled points represent Nov. 10, 2015, whose back-trajectory is shown in the previous figure.*

Most of the variation in column $CH_4$ and $N_2O$ is associated with the stratospheric circulation. Old airmasses from high latitude are depleted in $CH_4$ and $N_2O$. To remove these effects, and be able to more clearly see changes driven by the troposphere, $XCH_4$ is plotted versus $XN_2O$ which is similarly affected by stratospheric circulation, but not by tropospheric emissions.  This creates a correlation with the lower-left points representing high-latitude stratospheric airmasses and the upper right low-latitude airmasses.

The encircled points on Fig. C.2 were measured on Nov 10, 2015, when JPL was downwind of the Aliso Canyon USF leak.  The indicate $XCH_4$ enhancements of over 2%, which probably represent a 10+% enhancement in the PBL with no enhancement above. There is also a general tendency for higher $CH_4$ values when $C_2H_6$ is elevated on other days too, as seen from the dark blue points (low $C_2H_6$) being predominantly in the lower right of the figure and the greener points (higher $C_2H_6$) being located toward the upper left.

**Code Availability**

The GFIT code used for the analysis of MkIV spectra is identical to that used by the TCCON project. It is publicly available under license from the California Institute of Technology for non-commercial use. It can be cloned from:

   hg clone https://parkfalls.gps.caltech.edu/tccon/stable/hg/ggg-stable/

after signing the license agreement and being issued a password.

**Data Availability**

The ground-based MkIV data used in this paper can be downloaded from two sites:

https://mark4sun.jpl.nasa.gov/ground.html

ftp://ftp.cpc.ncep.noaa.gov/ndacc/station/barcroft/ames/ftir/

**Authors Contributions**

Toon, Sung, Blavier for data acquisition. Toon and Yu for data interpretation.

**Competing Interests**

No competing interests.

**Acknowledgements**

The authors gratefully acknowledge the NOAA Air Resources Laboratory (ARL) for the provision of the HYSPLIT transport and dispersion model and/or READY website (https://www.ready.noaa.gov) used in this publication. We

Deleted: *C*

Deleted: C

Deleted: K

thank NCEP and GEOS FPIT for their atmospheric analyses. We also acknowledge the NOAA ESRL GMD for distributing in situ data of $C_3H_8$ and $C_2H_6$. We thank NASAs Upper Atmosphere Composition Observation (UACO) program for funding support.

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

> **Formatted:** Font: Times New Roman, 10 pt
>
> **Formatted:** Font: 10 pt

**Page 5: [1] Deleted**          **Microsoft Office User**          **2/9/21 9:46:00 AM**

**Page 22: [2] Deleted**         **Microsoft Office User**          **3/13/21 3:33:00 PM**

**Page 23: [3] Deleted**         **Microsoft Office User**          **2/4/21 4:24:00 PM**