# Peer review of "Spectrometric measurements of atmospheric propane (C3H8)"

_Atmospheric Chemistry and Physics, 2020_

## Referee Comment (RC1) · Anonymous Referee #1 · 3 Jan 2021

The paper decribes the first spectroscopic measurements of atmospheric propane (C3H8) using the solar absorption spectrometry. The paper is very well written, very interesting, and should be published. ACP is the right journal.

Major comments:

The paper states that 5000 ground-based spectra have been recorded with the Mark4 at 1200 days, and 12 different sites (lines 59-60). Figure 2 gives a result for the average of all 5000 spectral fits. I assume that these 5000 spectra have been recorded at a variety of many different solar zenith angles. Also, I assume the figure gives the average of the retrievals for spectra recorded at very different altitudes, from the ground up to 2.5 km altitude. I do not see that its possible to average the results of these very different retrievals. This might be possible, but requires an explanation. Or, is it possible

to show the result for one spectral fit?

The paper discusses in several Figures the correlation between C3H8 and C2H6. When I look at Figure 2, it seems that both trace gases absorb in the same spectral region. The paper mentions ... litte spectroscopic "cross talk" between these two gases (line 109) but it might be useful to discuss this a bit more quantitatively, if possible. I accept that the absorption features of both trace gases are very different, but a small discussion would be helpful.

The comparison between in-situ and total column data is difficult. However, using the assumed a-priori profile used, together with the scaling factor for the retrievals, it is possible to get a rough idea on the surface concentration. This would be useful for the comparison. It could show a good agreement, or that the a-priori assumption is wrong (assumed that the spectroscopy is correct).

Minor comments: The absorption of C3H8 is quite weak, and the retrieval has been performed very carefully and at its limits. It would be good to have a small discussion about the possibility to measure C3H8 at other NDACC sites. Does it make sense that other groups look at C3H8 in their spectra?

Figures 3, 4 and 6 contain a lot of information, but since data from different sites are in one Figure, the reader might get the impression that the overall scatter is extremely high, and that the measurements are useless. I suggest to start with Figure 10, which gives a clear and nice indication of the measurements and their limits. The other results might also be presented differently, may be showing the results for the different sites individually.

For Figure 11, left it might make sense to plot annual averages instead of the individual data. This holds also for several other Figures.

Line 37: ... essentially zero at ... What is essentially zero? Please give a concentration, or upper limit.

Line 143-146: As far as I understand, its written: At lower altitudes the uncertanties are quite high. In the next sentence its written: This allows to detect C3H8 under polluted conditions. I dont understand these two sentences, for me this is contradictory.

Line 148: 'use' should be 'us'

Line 203-209: I dont see why it is important to discuss CH4. For me this chapter could be deleted.

Line 266: The estimation given in this chapter is interesting, but I have the feeling that the 4%, given in line 266 is not an assumption, but a result, giving meaningful data. This chapter might be rewritten.

Line 278: I would suggest to include the averaging kernel figures in the main text.

---

## Referee Comment (RC2) · Debra Wunch (Referee) · 6 Jan 2021

This paper describes the retrieval results from a newly developed propane spectroscopic pseudo-line-list. The paper convincingly argues that the retrievals of propane are realistic (albeit a bit too high), and that the variability is sensible. This new line list will be very interesting to apply to spectra already collected by the NDACC and TCCON networks. The paper is well-written and is appropriate for publication in ACP. I do not have major comments, but several Minor/Technical comments:

Minor/Technical Comments

L12-13: This last sentence seems to be missing a few words for context. Maybe: "From high-altitude balloon borne MkIV solar occultation measurements, C3H8 was

not detected at any altitude (5-30km\*) in any of its 25 flights." (Please replace \* with the correct numbers.)

L26-27: I had to re-read this last sentence a few times, because the previous sentence begins with "In contrast," referring to the ethane lifetime. I think just switching the 2nd and 3rd sentences in this paragraph would help make your point clearer: that the 2-8-week lifetime is long enough to affect a large fraction of the hemisphere.

L45: add "period" to: ". . . in the 2005-2010 \*period\* based on. . ."

L85: remove "then". It would be helpful to briefly explain why the 1400 cm-1 band would be better on cold planets for thermal emission spectrometry.

Fig 1.: In my copy, this figure is blurry, and the text is too small for me to read clearly.

L97: I cannot download the report that's linked to this address; it gives the message that the requested URL was not found.

L143-146: I think these sentences are missing a statement that although the absolute uncertainties are larger, the absolute values of the total columns themselves are much larger, so the relative uncertainties are smaller. (If that's indeed true.)

L148: use -> us

L150: Are the retrievals of C2H6 you're correlating with C3H8 done in the same window as the C3H8, or are they done in an independent window? (You answer this question later on L214, but it may be worth clarifying that the C2H6 retrievals are in independent windows here as well.)

L153: It would be helpful to introduce the "X"C3H8 notation at this point, as it is used in the figure and the next paragraph.

L166: measurement\*s\* (add an "s")

L181: I suggest combining the sentence beginning with "And" to the previous sentence.

[Figure]

Fig. 6: Why can't I see error bars on XC2H6? I can clearly see them for XC3H8 and XCO. (You state later on L210-211 that the ethane error bars are small – so small that they are the same size as the points on the figure? If so, I'm surprised they are so much smaller than XCO errors.)

L205: Could you use the global methane growth rate, or the Mauna Loa methane growth rate to detrend the XCH4? Or, instead, use an anomaly analysis (i.e., subtract the minimum or median total column of each gas on each day before plotting correlations of the various gases)? The anomaly analysis may help improve correlation coefficients and could help interpret the results.

Fig. 9b: I could not access the provided link to see the temporarily removed figure. Has permission been granted?

L362: produce -> product

L363: show ** increasing (no "a")

L379: I assume you mean Wunch et al., 2011.

---

## Referee Comment (RC3) · Anonymous Referee #3 · 15 Jan 2021

This paper describes the first dataset of atmospheric propane (C3H8) retrieved from remote sensing measurements. A large amount of data obtained at 12 different ground-stations over several years is presented. The authors give convincing explanations for the enhanced propane amounts observed at two sites, namely losses during exploitation of natural gas and pollution by liquified petroleum gas in large cities. They substantiate their conclusions by correlation with C2H6 and with CO, tracers for gas losses and anthropogenic pollution. The manuscript is well structured and its objective is clear. Therefore I recommend publication in ACP.

Major comments:

The strong correlation with C2H6 for, e.g., the Ft. Sumner data is convincing evidence that the C3H8 VMRs are real, maybe with an unknown bias. Nevertheless I would be

[Figure]

interested in a plot similar to Figure 2 but for retrievals using all other fit-parameters except of C3H8 to see, if the residuals become significantly larger. Maybe this could be shown in the Appendix. A table containing the name, acronym, geographical coordinate, altitude and country of each of the 12 sites at the beginning of Section 3 would be helpful.

Figures 3-5: The authors state that Fig. 3 contains C3H8 column amounts from 12 sites. If so, the sites should be made better distinguishable. I can detect 9 different lines or 7 different colours in the upper panel at the most. The same applies for Figures 4 and 5. Especially the two different "greens" are not easily to distinguish. Maybe it would help if one "green" was a bit darker? I concede that it might hardly be possible to present the data using 12 clearly distinguishable colours. Maybe subdivision into low-and high-altitude stations would help, by which the lines in the upper panel of Fig. 3 would be easier to distinguish as well. Further, every station should be listed with its associated colour code in the figure captions. Which station is, e.g., blue (0 km)?

L260ff: I would appreciate a little more information how the authors derive 0.72E+19 molecules cm-2 in total and 3E+16 molecules cm-2 of propane from 15 billion cu. ft. produced per day.

Minor comments:

L10: Aren't rather the high C3h8 amounts than the variations "correlated" with back trajectories from SE New Mexico ...?

L55: "the entire 650–5650 cm-1 range" instead of "the entire 650–5650 cm-1"

L63/64: "(Irion et al., 2002)" instead of "(Irion et al., 2003)"

L71: I do only count 9 simultaneously-fitted scalars. Can you help me?

L75: The acronym TCCON should be introduced in line 65. Further, what is GGG2014?

L78: What is GGG2020?

[Figure]

L79: I think "less than 10%" is more appropriate than "less than 10% rms", because specifications in percent are dimensionless. Further, "shown in Fig. A2" more specifically indicates, where the differences can be seen, than "shown in appendix A".

L82: Which infra-red lab?

Figure 1: The labelling of the axes and the legend is rather small and blurred and should be represented clearer.

L124: Here it says: "Table Mountain Facility at 2.2 km" but in the captions of Fig. 3 "orange=2.25 km (TMF)". Please adjust the heights.

L130: I believe the sentence "So C3H8 has only ..." would become clearer, if it would be exchanged with the preceding sentence.

Figure 2: The colors for H2O and HDO are hardly to distinguish.

L155: The upper row and not the left hand panels show XC3H8. Please change into "The upper row of Fig.4 shows the XC3H8 time series plotted versus year (left) and versus day of the year (right)."

Figure 4, captions: It should be mentioned that the colour coding is the same as in Fig. 3. This information could then be removed in lines 170/171.

L166: "The Antarctic measurement (blue) are even lower than they appear because ..." sounds strange. What about "The Antarctic measurements (blue) (of C2H6?) are very low (0.2-0.3 ppb) and most probably even lower during the rest of the year, because ..."?

L172: The sentence "In fact, the highest VMRs of C2H6 were seen from there, even more than from JPL ..." seems to be contradictionary to L167/168: "The highest ever C2H6 was measured from JPL (cyan) in late 2015 ..."

Figure 5: "... by site altitude like in Fig. 3." (?)

L197: The authors state: "... but only when the wind direction is from the SE quadrant (green/lime colors)." On the other hand it says green = 180 deg, lime = 220 deg in Fig. 6. Shouldn't the SE quadrant be in the direction 135+-45 deg? Or is the wind direction not counted clockwise, beginning in the north?

Figure 7: Do the authors apply a standard regression analysis (minimization of the squared vertical distances)? If so, wouldn't it be better to correlate XC3H8 versus XC2H6, because the X2H6-errors are much smaller?

L228: Why was a trajectory altitude of 0.4 km over Ft. Sumner selected?

L280: Why is 25% low only half the problem? Because of the rest of the profiles above the PBL or due to other reasons? Please explain.

Figure B.2: The axes-units are missing.

Technical comments:

L9: "shows" instead of "show"?

L16: "losses" instead of "loses"?

L32: "is therefore is": one "is" should be removed

L40: "show a large" instead of "show large a"

L65/66: "(Toon et al., 2016; 2018a; 2018b)" instead of "Toon, 2016; 2018a; 2018b)"

L124: "in red" and "in orange" instead of "= Red" and "= Orange"

L166: "measurements" instead of "measurement"

L363: "shows" instead of "show a"

L364: "when back-trajectories from SE New Mexico and West Texas ...": I think a verb is missing here.

---

## Author Comment (AC1) · 15 Mar 2021

Referee comments in black. *My responses in green italics.*

**Anonymous Referee #1**

The paper decribes the first spectroscopic measurements of atmospheric propane (C3H8) using the solar absorption spectrometry. The paper is very well written, very interesting, and should be published. ACP is the right journal. *Thank you!*

The paper states that 5000 ground-based spectra have been recorded with the Mark4 at 1200 days, and 12 different sites (lines 59-60). Figure 2 gives a result for the average of all 5000 spectral fits. I assume that these 5000 spectra have been recorded at a variety of many different solar zenith angles. Also, I assume the figure gives the average of the retrievals for spectra recorded at very different altitudes, from the ground up to 2.5 km altitude. I do not see that its possible to average the results of these very different retrievals. This might be possible, but requires an explanation. Or, is it possible to show the result for one spectral fit?

*We agree that due to the non-linearity of transmittance with airmass, the effect of averaging spectral fits taken at different SZA or altitudes does not give the same result as would be obtained at the average SZA and average altitude, especially for strong absorption lines. So it would not be valid to fit an average spectrum. But since we fitted individual spectra and then averaged, the resulting spectral fit provides a valid representation of the residuals of the overall dataset, with the advantages of reducing the noise level and some systematic errors, e.g., due to atmospheric temperature errors. If we were to use a single spectrum fit, which site, zenith angle, and propane amount would we choose? Whatever spectrum I choose, it would be representative of only a small fraction the dataset.*

The paper discusses in several Figures the correlation between C3H8 and C2H6. When I look at Figure 2, it seems that both trace gases absorb in the same spectral region. The paper mentions ... litte spectroscopic "cross talk" between these two gases (line 109) but it might be useful to discuss this a bit more quantitatively, if possible. I accept that the absorption features of both trace gases are very different, but a small discussion would be helpful.

[Figure]

*The following new material is now included in the paper as Appendix A. We compute Pearson Correlation Coefficients (PCC) from the a posteriori covariance matrix for each of the 5000 spectral fits. The upper panel (left) shows the PCC between retrieved $C_3H_8$ and $C_2H_6$ for the 5000 spectra. Points are plotted versus year with the same site-altitude-dependent coloring as in Fig.3. The PCC between $C_3H_8$ and $C_2H_6$ averages about -0.7, which means that they are fairly strongly anti-correlated. This is due to their overlapping absorption features at 2967.5 $cm^{-1}$. So as retrieved $C_2H_6$ increases, $C_3H_8$ will decrease, and vice versa. The PCCs are closer to zero for the high-altitude sites (red & orange), presumably due to the reduced pressure broadening and decreased $H_2O$ absorption causing the $C_2H_2$ and $C_2H_6$ absorption spectra to become more distinct. This anti-correlation could be reduced by use of a wider window to introduce additional $C_2H_6$ features that don't overlap the $C_3H_8$ Q-branch, but this would likely also encompass large residuals without adding any $C_3H_8$ information.*

*Also shown (lower panel) is the correlation between $C_3H_8$ and the fitted Continuum Level (CL). These have a correlation of about +0.65 at low SZA, decreasing at higher SZA. So the more $C_3H_8$ that is retrieved, the higher the continuum level has to be to match the measured spectrum, due to the fact that the $C_3H_8$ absorption spectrum has a broad continuum-like component. The PCCs between $C_3H_8$ and the other retrieved parameters (e.g., $H_2O$, HDO, $CH_4$, Continuum Tilt, Frequency Shifts) were all much closer to zero than with $C_2H_6$ and CL.*

*The high PCC between $C_3H_8$ and $C_2H_6$ doesn't necessarily imply a large uncertainty in the $C_3H_8$. It just means that the large component of the $C_2H_6$ uncertainty gets projected onto the $C_3H_8$. Ditto for the CL. But provided the $C_2H_6$ and CL are well retrieved, their effect on the $C_3H_8$ will not dominate.*

The comparison between in-situ and total column data is difficult. However, using the assumed a-priori profile used, together with the scaling factor for the retrievals, it is possible to get a rough idea on the surface concentration. This would be useful for the comparison. It could show a good agreement, or that the a-priori assumption is wrong (assumed that the spectroscopy is correct).

*Agreed. But the retrieval returns just one piece of information on $C_3H_8$. So the deduced surface concentration would be highly dependent on the assumed a priori, which is highly uncertain under conditions of enhanced $C_3H_8$.*

Minor comments: The absorption of C3H8 is quite weak, and the retrieval has been performed very carefully and at its limits. It would be good to have a small discussion about the possibility to measure C3H8 at other NDACC sites. Does it make sense that other groups look at C3H8 in their spectra?

*Yes, NDACC sites in polluted locations should look for $C_3H_8$.  For sites in clean locations, however, the spectroscopy needs further improvement to be able to obtain useful measurements under background C3H8 levels. The strong H2O line at 2966.0 $cm^{-1}$ is particularly troublesome in driving up the fitting residuals. We have modified the summary section to further emphasize this point.*

Figures 3, 4 and 6 contain a lot of information, but since data from different sites are in one Figure, the reader might get the impression that the overall scatter is extremely high, and that the measurements are useless. I suggest to start with Figure 10, which gives a clear and nice indication of the measurements and their limits. The other results might also be presented differently, may be showing the results for the different sites individually.

*Figure 10 contains just the JPL data, one of 12 observation sites. And it plots Xgas values. I continue to prefer to start with an overview of the whole dataset, show the raw column data, then explain how to derive Xgas values from the columns, then look at individual sites. Yes, some readers might initially think that the measurements are useless, but those that persevere will discover that C3H8 is highly variable.*

For Figure 11, left it might make sense to plot annual averages instead of the individual data. This holds also for several other Figures.

*Not sure what is gained by plotting annual averages? Sure, it will reduce clutter in the SGP panels, but information in the variability is lost.*

Line 37: ... essentially zero at ... What is essentially zero? Please give a concentration, or upper limit.

*Replaced "essentially zero" with "below 50 ppt".*

Line 143-146: As far as I understand, its written: At lower altitudes the uncertainties are quite high. In the next sentence its written: This allows to detect C3H8 under polluted conditions. I dont understand these two sentences, for me this is contradictory.

*Good point. Sentence has been reworded as " But the C3H8 increases far more, allowing $C_3H_8$ to be detected at these low-altitude sites under polluted conditions, despite the poorer absolute uncertainties."*

Line 148: 'use' should be 'us'  *Fixed.*

Line 203-209: I dont see why it is important to discuss CH4. For me this chapter could be deleted.

*I agree that CH4 is not the main thrust of this paper. But CH4 is the main constituent of NG and is well-measured by the MkIV, so it would seem remiss, or even evasive, not to show some MkIV CH4 data and explain their limitations toward diagnosing the causes of the C3H8 enhancement.*

Line 266: The estimation given in this chapter is interesting, but I have the feeling that the 4%, given in line 266 is not an assumption, but a result, giving meaningful data. This chapter might be rewritten.

*Yes, you are right. I have re-written this paragraph. I have also realized that the Permian Basin Iol and NG field is larger than I originally estimated.  This is also corrected.*

Line 278: I would suggest to include the averaging kernel figures in the main text.

*Done. A new section was created "3.1 Averaging Kernels".*

---

## Author Comment (AC2) · 15 Mar 2021

Referee comments in black. *My responses in green italics.*

**Debra Wunch (Referee #2)**

This paper describes the retrieval results from a newly developed propane spectroscopic pseudo-line-list. The paper convincingly argues that the retrievals of propane are realistic (albeit a bit too high), and that the variability is sensible. This new line list will be very interesting to apply to spectra already collected by the NDACC and TCCON networks. The paper is well-written and is appropriate for publication in ACP. I do not have major comments, but several Minor/Technical comments: *Thank you!*

Minor/Technical Comments

L12-13: This last sentence seems to be missing a few words for context. Maybe: "From high-altitude balloon borne MkIV solar occultation measurements, C3H8 was not detected at any altitude (5-30km*) in any of its 25 flights." (Please replace * with the correct numbers.) *Fixed.*

L26-27: I had to re-read this last sentence a few times, because the previous sentence begins with "In contrast," referring to the ethane lifetime. I think just switching the 2nd and 3rd sentences in this paragraph would help make your point clearer: that the 2-8- week lifetime is long enough to affect a large fraction of the hemisphere. *Moved second sentence (about C2H6) to 4 paragraphs later.*

L45: add "period" to: "... in the 2005-2010 *period* based on..." *Done.*

L85: remove "then". It would be helpful to briefly explain why the 1400 cm-1 band would be better on cold planets for thermal emission spectrometry. *Done both.*

Fig 1.: In my copy, this figure is blurry, and the text is too small for me to read clearly.

*Fig.1 looks crystal-clear in my MSWord file, but is indeed fuzzy in the PDF that I created using "Preview". I \*think\* that the final paper is submitted in the form of a MSWord file, not a PDF, in which case ACP will hopefully have a better PDF converter than I do.*

L97: I cannot download the report that's linked to this address; it gives the message that the requested URL was not found. L143-146: I think these sentences are missing a statement that although the absolute uncertainties are larger, the absolute values of the total columns themselves are much larger, so the relative uncertainties are smaller. (If that's indeed true.) *Fixed.*

L148: use -> us *Fixed.*

L150: Are the retrievals of C2H6 you're correlating with C3H8 done in the same window as the C3H8, or are they done in an independent window? (You answer this question later on L214, but it may be worth clarifying that the C2H6 retrievals are in independent windows here as well.) *Fixed.*

L153: It would be helpful to introduce the "X"C3H8 notation at this point, as it is used in the figure and the next paragraph. *Fixed.*

L166: measurement*s* (add an "s") *Fixed.*

L181: I suggest combining the sentence beginning with "And" to the previous sentence. Fixed.

Fig. 6: Why can't I see error bars on XC2H6? I can clearly see them for XC3H8 and XCO. (You state later on L210-211 that the ethane error bars are small – so small that they are the same size as the points on the figure? If so, I'm surprised they are so much smaller than XCO errors.)

*You can see error bars in the upper right, where the XC3H8 values are high, but not for C3H8 values below 2 ppb. The diamond symbols have a height of nearly 0.1 ppb, so an error bar will only be discernable if it exceeds 0.05 ppb which is 5% at 1ppb. XCO errors are about 5% but are much more easily seen due to the much smaller dynamic range of the CO data. So if I were to zoom the y-scale to cover only 0.5 to 1.5 ppb, using the same panel size, then the error bars would grow by a factor 3.5 whereas the diamond symbols would stay the same size in absolute terms. So the error bars would all become visible.*

L205: Could you use the global methane growth rate, or the Mauna Loa methane growth rate to detrend the XCH4? Or, instead, use an anomaly analysis (i.e., subtract the minimum or median total column of each gas on each day before plotting correlations of the various gases)? The anomaly analysis may help improve correlation coefficients and could help interpret the results.

*For a gas that increased linearly with time (e.g., N2O) it would be easy to detrend the MkIV data. But CH4 has a growth that surges and then levels off. So I think that I would need information about the age of the air that was sampled in each observation at each site in order to do a proper detrending. Regarding correlating deviations from the daily mean, remember that the MkIV only observes for 1-2 hours each day, in general, so the measured columns don't have much opportunity to change.*

Fig. 9b: I could not access the provided link to see the temporarily removed figure. Has permission been granted?

Link works for me. Can you see the link below. It is a different link to the same figure, but without the accompanying article: https://www.spglobal.com/platts/plattscontent/ assets/ images/latest-news/20191219-rig-count.jpg. Permission has not yet been granted after 4 months, so I've given up.

L362: produce -> product Done
L363: show ** increasing (no "a") Done
L379: I assume you mean Wunch et al., 2011. Yes. Fixed.

---

## Author Comment (AC3) · 15 Mar 2021

Referee comments in black. *My responses in green italics.*

**Anonymous Referee #3**

This paper describes the first dataset of atmospheric propane (C3H8) retrieved from remote sensing measurements. A large amount of data obtained at 12 different ground- stations over several years is presented. The authors give convincing explanations for the enhanced propane amounts observed at two sites, namely losses during exploitation of natural gas and pollution by liquified petroleum gas in large cities. They substantiate their conclusions by correlation with C2H6 and with CO, tracers for gas losses and anthropogenic pollution. The manuscript is well structured and its objective is clear. Therefore I recommend publication in ACP. *Thank you!*

Major comments:

The strong correlation with C2H6 for, e.g., the Ft. Sumner data is convincing evidence that the C3H8 VMRs are real, maybe with an unknown bias. Nevertheless, I would be interested in a plot similar to Figure 2 but for retrievals using all other fit-parameters except of C3H8 to see, if the residuals become significantly larger. Maybe this could be shown in the Appendix.

*Good idea. Fig.2 has been expanded to include an extra panel showing the residuals that would arise if there were no atmospheric C3H8. Residuals increase from 0.36% to 0.39%*

A table containing the name, acronym, geographical coordinate, altitude and country of each of the 12 sites at the beginning of Section 3 would be helpful. *Done.*

Figures 3-5: The authors state that Fig. 3 contains C3H8 column amounts from 12 sites. If so, the sites should be made better distinguishable. I can detect 9 different lines or 7 different colours in the upper panel at the most. The same applies for Figures 4 and 5. Especially the two different "greens" are not easily to distinguish. Maybe it would help if one "green" was a bit darker? I concede that it might hardly be possible to present the data using 12 clearly distinguishable colours. Maybe subdivision into low-and high-altitude stations would help, by which the lines in the upper panel of Fig. 3 would be easier to distinguish as well. Further, every station should be listed with its associated colour code in the figure captions. Which station is, e.g., blue (0 km)?

*Yes, 12 colors are too many to be distinguishable. The easiest solution is to drop the four sites with fewest observations, namely JPL Mesa, CA, Palestine, TX, and Mountain View, CA, and Lynn Lake, Manitoba. This significantly reduces the color ambiguity, without noticeable changes to the figure. Terse sites contributes only 1% of the total observations.*

L260ff: I would appreciate a little more information how the authors derive 0.72E+19 molecules cm-2 in total and 3E+16 molecules cm-2 of propane from 15 billion cu. ft. produced per day.

*I expanded the text as follows. "The Permian basin currently produces 16 billion cu.ft./day of NG (https://www.eia.gov/petroleum/drilling/pdf/permian.pdf) over an area of 220,000 km². The molar volume of an ideal gas at STP is 22.4 liters. One cu. ft. is 28.3 liters. So 16 billion cu. ft. is 20 billion moles of NG or $120 \times 10^{32}$ molecules per day. Over an area of 220,000 km² or $2.2 \times 10^{15}$ cm², this represents an average areal production of $55 \times 10^{17}$ molec./cm²/day. Assuming that the Permian basin of 480 km long, at an average low-level wind speed of 15 km/hour, an air parcel will take 32 hours (1.33 days) to traverse the Basin, during which time $73 \times 10^{17}$ molecules/cm² will have been extracted. Of this, 10% will be $C_3H_8$ (Howard et al., 2015), so if all this production were released into the atmosphere we would expect a $C_3H_8$ column enhancement of $73 \times 10^{16}$.*

*In airmasses with trajectories from the SE, we see maximum $C_3H_8$ column enhancements of only $3 \times 10^{16}$ molecules/cm², which suggests that only 4% of the NG escapes into the atmosphere and that 96% of the NG is successfully captured (or burnt by flaring). Of course, this analysis assumes that the Permian basin is a uniform emitter and that the back trajectory wind speeds are accurate. There are likely hot spots with higher-than-average emissions, and regions with little NG production."*

Minor comments:

L10: Aren't rather the high C3h8 amounts than the variations "correlated" with back trajectories from SE New Mexico ...? *Agreed and fixed.*

L55: "the entire 650–5650 cm-1 range" instead of "the entire 650–5650 cm-1" Fixed.

L63/64: "(Irion et al., 2002)" instead of "(Irion et al., 2003)" *Fixed.*

L71: I do only count 9 simultaneously-fitted scalars. Can you help me?

*We retrieve 5 gases ($C_3H_8$, $H_2O$, $CH_4$, $C_2H_6$, HDO), two frequency stretches (solar & telluric), two continuum parameters (a straight line is defined by two coefficients), and a zero-level offset.*

L75: The acronym TCCON should be introduced in line 65. *Done.*

Further, what is GGG2014? L78:

*Added sentence "The entire package including spectral fitting software, spectroscopic linelists, and software to generate a priori VMR/T/P profiles, is denoted GGG."*

What is GGG2020?

*Added "an updated version of GGG with improved a priori VMR/T/P profiles, spectroscopy, and software (Laughner et al., 2021)"*

L79: I think "less than 10%" is more appropriate than "less than 10% rms", because specifications in percent are dimensionless.  *Agreed. Removed "rms".*

Further, "shown in Fig. A2" more specifically indicates, where the differences can be seen, than "shown in appendix A". *Done.*

L82: Which infra-red lab? *Added "Pacific North-West National Laboratory (PNNL)" and also a reference to Sharpe et al. (2004)*

Figure 1: The labelling of the axes and the legend is rather small and blurred and should be represented clearer. *Done.*

L124: Here it says: "Table Mountain Facility at 2.2 km" but in the captions of Fig. 3 "orange=2.25 km (TMF)". Please adjust the heights.  *All are now 2.26 km.*

L130: I believe the sentence "So C3H8 has only ..." would become clearer, if it would be exchanged with the preceding sentence.  *Agreed. Done.*

Figure 2: The colors for H2O and HDO are hardly to distinguish.  *Agreed.  I have re-made the figure with more color separation between H2O and HDO.*

L155: The upper row and not the left hand panels show XC3H8. Please change into "The upper row of Fig.4 shows the XC3H8 time series plotted versus year (left) and versus day of the year (right)."

*Changed to "The upper and lower rows of Fig.4 show the XC3H8 and XC2H6 time series, respectively, plotted versus year (left) and versus day of the year (right)."*

Figure 4, captions: It should be mentioned that the colour coding is the same as in Fig. 3. This information could then be removed in lines 170/171. *Done.*

L166: "The Antarctic measurement (blue) are even lower than they appear because ..." sounds strange. What about "The Antarctic measurements (blue) (of C2H6?) are very low (0.2-0.3 ppb) and most probably even lower during the rest of the year, because ..."? *Agreed and fixed.*

L172: The sentence "In fact, the highest VMRs of C2H6 were seen from there, even more than from JPL ..." seems to be contradictionary to L167/168: "The highest ever C2H6 was measured from JPL (cyan) in late 2015 ..." *Agreed. The latter sentence now says "The highest C2H6 ever measured from JPL (cyan) was in late 2015 ..."*

Figure 5: "... by site altitude like in Fig. 3." (?) *Yes. Fixed.*

L197: The authors state: "... but only when the wind direction is from the SE quadrant (green/lime colors)." On the other hand it says green = 180 deg, lime = 220 deg in Fig. 6. Shouldn't the SE quadrant be in the direction 135+-45 deg? Or is the wind direction not counted clockwise, beginning in the north?

*This is a simple typo: lime was written instead of cyan. The text now states ""... but only when the wind direction is from the SE quadrant (green/cyan) colors". Lime is now listed with the other wind directions in the next sentence.*

Figure 7: Do the authors apply a standard regression analysis (minimization of the squared vertical distances)? If so, wouldn't it be better to correlate XC3H8 versus XC2H6, because the X2H6-errors are much smaller?

*Given the measured data [$x_i \pm ex_i, y_i \pm ey_i$] the regression minimizes*

$$\chi^2 = \Sigma_i (x_i - x')^2 / ex_i^2 + \Sigma_i (y_i - y')^2 / ey_i^2$$

*with respect to A and B, where $y' = A + Bx'$ is the fitted straight line. This is equivalent to the method described by (York, 1969; Wehr and Saleska, 2017) but assuming no correlation between the x- and y-uncertainties. So the error bars of both C2H6 and C3H8 are taken into account. Upon switching X and Y, the same the exact same PCC is obtained and the gradient simply reciprocates.*

L228: Why was a trajectory altitude of 0.4 km over Ft. Sumner selected?

*I thought that this was the altitude with the most transport of short-lived gas molecules released at the surface. At lower altitudes there is more C3H8, but the wind speed is less, especially at night. At higher altitudes above the PBL the winds are stronger but the gas concentration drops off rapidly.*

L280: Why is 25% low only half the problem? Because of the rest of the profiles above the PBL or due to other reasons? Please explain.

*Paragraph has been re-written as " A puzzle in our findings is that when both $C_3H_8$ and $C_2H_6$ are elevated, we measure 22% more $C_3H_8$ than $C_2H_6$ (see fig.9). Yet independent essays of well-head wet NG find 33% more $C_2H_6$ than $C_3H_8$ in the Permian basin (Howard et al., 2015). So we have a 55% discrepancy. We note that the $C_2H_6$ averaging kernel is 0.7 at the surface versus 0.9 for $C_3H_8$ (see Appendix B). So when these gases exceed their priors in the PBL, which is likely at high enhancements, both will be under-estimated, but $C_2H_6$ more so than $C_3H_8$. So this effect would cause the $C_3H_8/C_2H_6$ ratio to be 28% high, which explains half the 55% problem."*

Figure B.2: The axes-units are missing. *Added units.*

Technical comments:

L9: "shows" instead of "show"? *Fixed.*

L16: "losses" instead of "loses"? *Fixed.*

L32: "is therefore is": one "is" should be removed *Fixed.*

L40: "show a large" instead of "show large a" *Fixed.*

L65/66: "(Toon et al., 2016; 2018a; 2018b)" instead of "Toon, 2016; 2018a; 2018b)" *Fixed.*

L124: "in red" and "in orange" instead of "= Red" and "= Orange" *Fixed.*

L166: "measurements" instead of "measurement" *Fixed.*

L363: "shows" instead of "show a" *Fixed.*

L364: "when back-trajectories from SE New Mexico and West Texas ...": I think a verb is missing here. *Fixed.*

---

## Author Comment (AC4) · 11 May 2021

I have fixed Referee #2's three minor additional comments.